# Hofbauer cell function in the term placenta associates with adult cardiovascular and depressive outcomes

Eamon Fitzgerald [1,2,3] ✉, Mojun Shen [4], Hannah Ee Juen Yong [4], Zihan Wang[3], Irina Pokhvisneva[3], Sachin Patel [3], Nicholas O'Toole[1,2,3], Shiao-Yng Chan [4,5], Yap Seng Chong[4,5], Helen Chen[6,7], Peter D. Gluckman [4,8], Jerry Chan[6,7], Patrick Kia Ming Lee [9] & Michael J. Meaney [1,3,4,5,9] ✉

Pathological placental inflammation increases the risk for several adult disorders, but these mediators are also expressed under homeostatic conditions, where their contribution to adult health outcomes is unknown. Here we define an inflammation-related expression signature, primarily expressed in Hofbauer cells of the term placenta and use expression quantitative trait loci to create a polygenic score (PGS) predictive of its expression. Using this PGS in the UK Biobank we conduct a phenome-wide association study, followed by Mendelian randomization and identify protective, sex-dependent effects of the placental module on cardiovascular and depressive outcomes. Genes differentially regulated by intra-amniotic infection and preterm birth are over-represented within the module. We also identify aspirin as a putative modulator of this inflammation-related signature. Our data support a model where disruption of placental Hofbauer cell function, due to preterm birth or prenatal infection, contributes to the increased risk of depression and cardiovascular disease observed in these individuals.

Early studies, foundational to the developmental origins of health and disease (DOHaD) hypothesis, described an increased risk of both cardiovascular disease and depression in those born at low birth weight[1,2]. As the site of maternal–fetal interface during pregnancy, the placenta is a critical regulator of these cardiovascular effects as well as those of mental health[3]. For instance, monozygotic twins that share a placenta are up to 6-fold more likely to be concordant for schizophrenia than monozygotic twins with separate placentas[4]. Genomic studies have further reinforced the essential role of the placenta in shaping adult health outcomes[5], with the role of placental inflammation being particularly compelling[6].

A large body of literature in model systems supports a causal role for prenatal infection in driving adverse adult behavioral outcomes[7], but the inflammatory mediators involved in the placental response to infection are also highly expressed under healthy conditions[8,9]. Indeed inflammation is a critical regulator of placental and fetal development, with the first trimester often conceptualized as pro-inflammatory and later periods as anti-inflammatory[10–12]. Inflammatory mediators are also critical for triggering uterine contractions and subsequent fetal delivery[11,12]. Furthermore, the loss of maternal decidual natural killer cells[13] or uterine dendritic cells[14] is sufficient for pregnancy termination in animal models. Work in the developing brain with microglia has also

[1]Sackler Program for Epigenetics and Psychobiology, McGill University, Montréal, Canada. [2]Ludmer Centre for Neuroinformatics and Mental Health, McGill University, Montréal, Canada. [3]Douglas Mental Health University Institute, Department of Psychiatry, McGill University, Montréal, Canada. [4]Singapore Institute for Clinical Sciences, Agency for Science, Technology & Research, Singapore, Singapore. [5]Yong Loo Lin School of Medicine, National University of Singapore, Singapore, Singapore. [6]KK Women's and Children's Hospital, Singapore, Singapore. [7]Duke-National University of Singapore, Singapore, Singapore. [8]The University of Auckland, Auckland, New Zealand. [9]Brain – Body Initiative, Agency for Science, Technology & Research, Singapore, Singapore. ✉e-mail: eamon.fitzgerald@mcgill.ca; michael.meaney@mcgill.ca

illustrated the important homeostatic role of inflammatory mediators, which traditionally were thought to subserve functions solely related to infection[15]. For instance, microglia are a tissue-specific macrophage and the placenta is home to its own tissue-specific macrophage, Hofbauer cells, which have important homeostatic roles in angiogenesis, placental remodeling and trophoblast maturation[16–18], while also displaying a robust response to infection[16]. However, little work has been done to identify the contribution of Hofbauer cells to adult health outcomes under non-pathogenic conditions. This is surprising as understanding homeostatic functions would also likely inform disease mechanisms.

In this study, we hypothesized inflammation-related patterns of placental gene expression would shape adult health outcomes in off-spring under non-pathogenic conditions. To test this hypothesis, we performed a series of experiments using the Singapore-based, Growing Up in Singapore Towards healthy Outcomes (GUSTO) and UK Biobank cohorts. We first used RNA sequencing from 42 placental villous samples, obtained as part of the GUSTO study, and used weighted correlation network analysis (WGCNA[19]) to identify an inflammation-related gene co-expression module, which was highly enriched in Hofbauer cells. We leveraged previously identified placental expression quantitative trait loci (eQTLs)[20] to generate a polygenic score (PGS; henceforth referred to as the fetoplacental PGS), which specifically predicted expression of the genes comprising this module. To explore the functional relevance of this fetoplacental PGS we conducted a phenome-wide association study (pheWAS) in the UK Biobank and identified significant associations (false discovery rate; FDR < 0.05) with 21 traits primarily within the cardiometabolic and mental health domains. We then used Mendelian randomization analyses to demonstrate protective sex-dependent effects on cardiovascular disease and depression-related outcomes. Next, we demonstrated that our placental module was highly enriched for genes differentially regulated by intra-amniotic infection, preterm birth, and in animal models of prenatal infection. We also describe a sharp decrease in expression of the cyan module in Hofbauer cells from pregnancies with an active SARS CoV-2 infection. These data support a model by which loss or disruption of Hofbauer function as a consequence of preterm birth or prenatal infection, respectively, contributes to the increased risk of depression and cardiovascular disease observed in these individuals[21,22]. Finally using the Drug-Gene Interaction database (DGIdb), we identify aspirin as a promising candidate that may have therapeutic value when used prophylactically in populations at high risk of intra-uterine infection.

## Results

### WGCNA in term placental villous samples

We used bulk RNA sequencing followed by WGCNA in 44 term placental villous samples obtained as part of the GUSTO study to identify inflammation-related gene expression patterns during non-pathogenic conditions. We sequenced samples consisting of Chinese (26 samples; 59%), Malay (7 samples; 16%), and Indian (11 samples; 25%) self-defined ethnicities, with 24 of the 44 sequenced samples being female. Clinical characteristics for sequenced placental samples are shown in Supplementary Data 1.

Following hierarchical clustering of the RNA sequencing data, 2 samples were removed as putative outliers (Supplementary Fig 1a) and 42 samples were submitted to WGCNA. From 31,097 expressed genes we identified 28 gene expression modules (Fig. 1a), ranging in size from 72 (skyblue module) to 6794 genes (turquoise module). Unassigned genes grouped into the gray module, with the constituents of each module comprehensively described in Supplementary Data 2.

### Characterization of inflammation-related modules

We first used gene ontology analysis to functional classify all modules, which identified five modules enriched for terms related to inflammation (Supplementary Data 3). We then built expression scores for each of the inflammation-related modules in a scRNA-seq dataset of term placental villous tissue (using only healthy control samples)[23, 24] (Fig. 1b and Supplementary Fig 2a). The cyan module had the strongest cell type specificity and was primarily expressed in Hofbauer cells, a placental macrophage localized to the villous[16] with important homeostatic[16,17,25,26] and pathogenic roles[16,27]. The cyan module was the largest inflammation-related module (486 genes) and also had the strongest preservation in an independent dataset[28] (Supplementary Fig 1c). Together with the increasingly appreciated role of tissue-specific macrophages in homeostasis[29], these characteristics made the cyan module the most attractive candidate for further study using a functional genomics approach.

Further characterization of the cyan module revealed it was also specifically enriched in Hofbauer cells from first-trimester scRNA-seq datasets[8,9], suggesting it may capture a persistent signature of Hofbauer cells (Supplementary Fig 2b and 2c). Several well-characterized inflammation-related genes were also present within the cyan module including members of the tumor necrosis factor (*TNFRSF11A, TNFAIP8L2, TNFRSF21, TNFSF13, TNFSF9*) and interleukin (*IL2RA, IL12RB2, IL1RL1*) families, as well as several genes associated with the myeloid lineage (*AIF1, CSF1, CD33, CD163*). Transcription factor enrichment analysis also identified targets of transcription factors associated with macrophage identity, including *SPI1* and *MAFB*, as enriched within the cyan module (Supplementary Fig 2d).

We next used a placental eQTL resource from Peng et al.[20] to identify SNPs contributing to variation in expression of cyan module genes (Supplementary Data 4 and 5). In a comparison with the Gene-Tissue Expression (GTEx) v8 eQTL catalog[30], the combination of these eQTLs associated with cyan module genes were highly specific to the placenta (Supplementary Fig 1d). We then identified these variants in the children of the GUSTO cohort and weighted them based on their effect size from Peng et al., before summing them within individuals to create a fetoplacental PGS (as described previously[31]). We reasoned that a PGS comprised of eQTLs would be representative of individual variation in cyan module expression. We validated this assumption using single sample gene set enrichment analysis in the GUSTO RNA-seq dataset ($p = 0.003$, $\beta = 0.02$, $n = 42$; Fig. 1c). A higher fetoplacental PGS was associated with increased expression of cyan module genes. The fetoplacental PGS did not predict the expression of the other WGCNA modules (Supplementary Data 6) and negative control PGS created in a similar fashion, using the WGCNA modules closest in size to the cyan module, had no effect on expression of cyan module genes (Fig. 1c; Supplementary Data 7). We provided additional controls for specificity by creating 2 further negative control PGS based on randomly generated lists of 486 genes (same size of the cyan module) from our sequencing dataset and another using the cyan module but weighting it with eQTLs from fetal cortical tissue[32]. The negative control PGS could not predict expression of the cyan module ($p = 0.11$, 0.61, and 0.19, respectively; Fig. 1c). Taken together these findings suggest the fetoplacental PGS has considerable specificity for genes comprising the module.

We also note that the fetoplacental PGS was not correlated with various perinatal factors including measures of maternal mental health, gestational age at birth, birth weight, offspring sex, and socioeconomic status (Supplementary Fig 1e).

Considering the importance of secreted factors during fetal development, we next investigated the association of the fetoplacental PGS with 27 cord blood molecules. These factors included cytokines (e.g. TNFα and IL6) and hormones (e.g. testosterone and insulin) with well-established prenatal effects. Using the fetoplacental PGS allowed us to expand our investigation to the entire sample of the GUSTO cohort with available genotype and cord blood data ($n = 194$–251 depending on molecule measured). Using the fetoplacental PGS as the predictor in a multiple regression analysis, the strongest effect was an association with monocyte chemoattractive protein 1 (MCP-1, also

known as CCL-2; $p = 0.005$, $\beta = 0.14$; Fig. 1d; Supplementary Data 8 and 9). Interestingly, Hofbauer cells secrete high levels of MCP-1[16], indicating our fetoplacental PGS may reflect Hofbauer cell function.

## Phenome-wide association study in the UK Biobank

There is a paucity of studies that have evaluated placental contributions to adult offspring outcomes in large human datasets. We therefore created a fetoplacental PGS in the UK Biobank, using identical criteria to the previously described fetoplacental PGS in the GUSTO cohort. We used this fetoplacental PGS to perform a pheWAS with 1831 traits as outcomes of interests. Only unrelated individuals were used and outcomes were categorized to an appropriate regression family using the PHESANT package[33]. We identified 21 significant associations (FDR $p$-value < 0.05; Fig. 2a and Supplementary Data 10) that were primarily related to anthropometric or mental health traits. All anthropometric traits had a positive direction of effect and all traits within the mental health domain had a negative direction of effect (Fig. 2a and Supplementary Data 10). When samples were split by sex, no association passed the threshold for multiple comparisons, but both anthropometric and mental health-related traits approached the

threshold for multiple comparisons in each sex (Supplementary Data 11 and 12).

We noted several of the pheWAS significant associations were risk factors for cardiometabolic (e.g. "hip circumference" and "trunk fat percent"[34]) or mental health disorders (e.g. "nervous feelings"[35]). The discovery of a relation between perinatal events and adult risk of cardiovascular disease and mental health disorders, such as major depressive disorder (MDD), has provided much of the foundation on which the DOHaD hypothesis has been built. Furthermore prenatal infection has been associated with an increased risk of both cardiovascular disorders[22] and MDD[21]. We did not find a correlation between our fetoplacental PGS and polygenic risk scores (PRS) for depression or cardiovascular disease (Supplementary Fig 3). Furthermore, the cyan module was not enriched in a MDD (beta=0.04, SE = 0.06, $P = 0.26$) or CAD (beta = −0.03, SE = 0.06, $P = 0.68$) GWAS.

## Mendelian randomization for cardiovascular and depressive outcomes

We next used Mendelian randomization as implemented in the TwosampleMR package[36,37]. We used the eQTLs that composed the

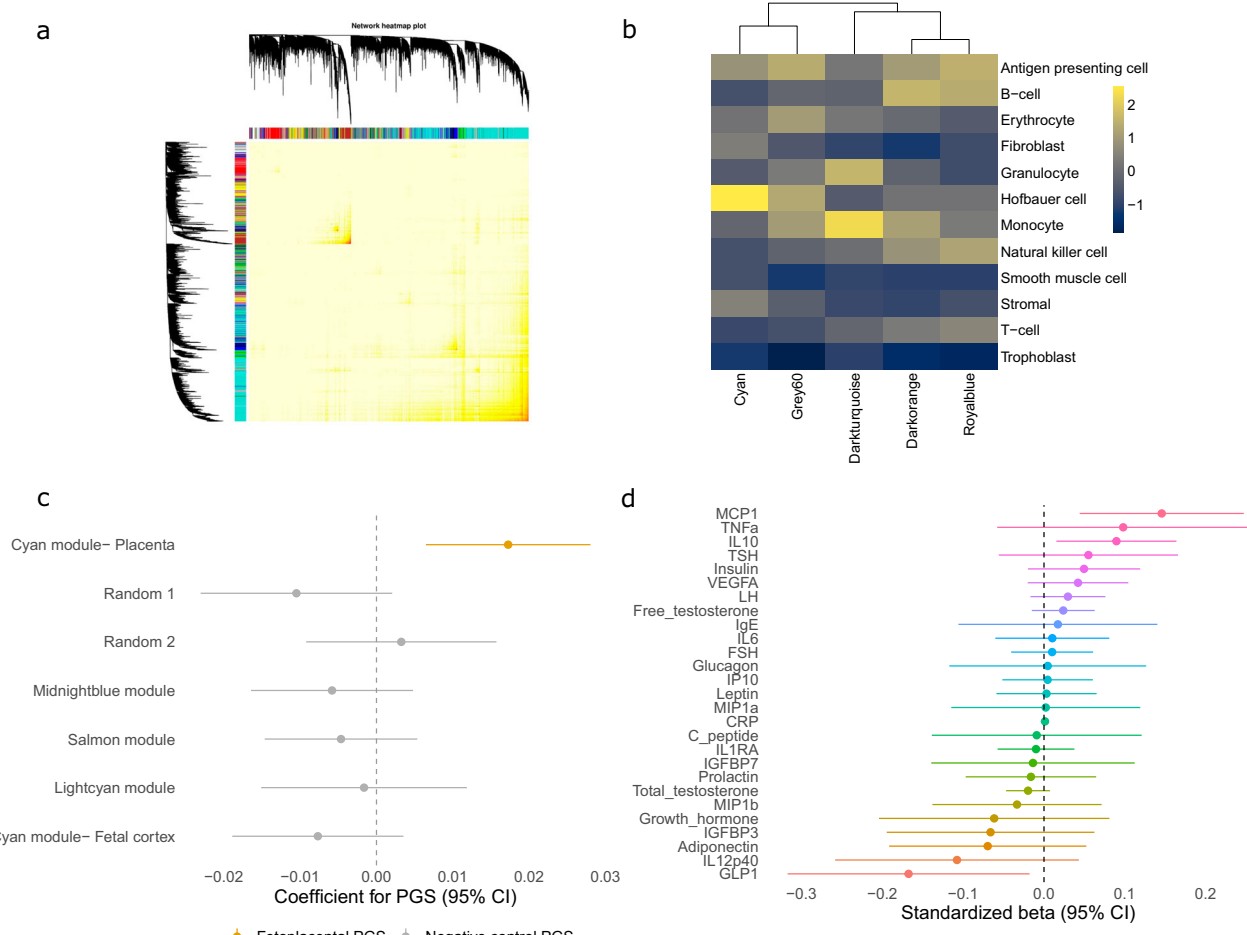

Fig. 1 | WGCNA of placental villous RNA sequencing data identifies an inflammation-related gene expression module that is highly enriched in Hofbauer cells. a Heatmap of the topological overlap matrix, with corresponding dendrogram and module assignment (represented as colors). b Expression of the five inflammation-related modules in scRNA-seq data from term control placental villi samples[23]. Each tile represents an expression score for each module within each cell type scaled by column, where yellow indicates higher relative expression and black lower relative expression. c Forest plot of the coefficient from multiple regression analyses with 95% confidence intervals, estimating the effect of the indicated PGS on single sample gene set

enrichment analysis scores from the cyan module of the GUSTO RNA-seq dataset. The first 3 genetic principal components and sex were used as covariates in the analysis ($n = 42$ samples with RNA-seq data). Data are presented as coefficient beta values ±95% confidence intervals. d Multiple regression analysis investigating the effect of the fetoplacental PGS on 27 molecules measured in cord blood. All outcomes were log transformed and scaled. The first three genetic principal components, sex and gestational age at birth were used as covariates in the analyses ($n = 194$–251 depending on molecule measured, precise sample sizes can be found in Supplementary Data 8). Data are presented as coefficient beta values ±95% confidence intervals.

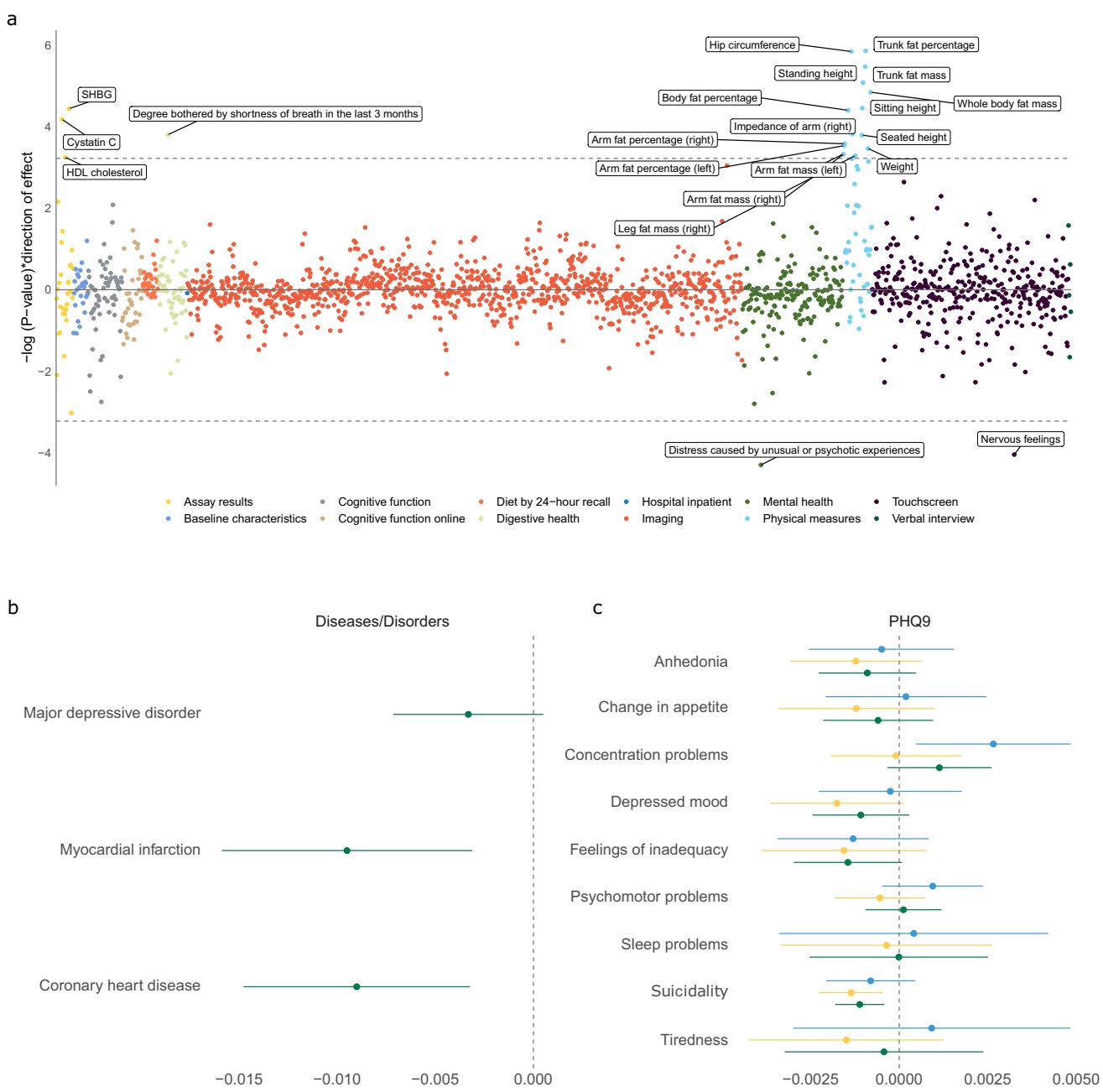

**Fig. 2 | Phenome-wide association study and mendelian randomization analyses identify associations between the cyan module and adult outcomes.**
**a** Miami plot showing the -log *p*-value corresponding to the effect of the fetoplacental PGS on 1831 outcomes (using two-sided regression models, as specified in the PHEASANT package) in the full UK Biobank sample (males and females combined) multiplied by the direction of effect. Dashed line indicates the FDR threshold for multiple comparisons, all values exceeding this threshold are labeled. Points are colored as per their categorizations in the UK Biobank database. In each regression the first 10 genetic principal components, age, sex, genotype array and assessment center (categorical variable) are used as covariates. In **b** and **c** Mendelian randomization is used with the inverse variance weighted (IVW) method to analyze the

effect of placental eQTLs on various adult outcomes. In both panels, green represents males and females combined, yellow represents females only and blue represents males only. Data are presented as the IVW estimate ±95% confidence intervals. The dashed line represents 0 in both panels. All tests were 2-sided. In **b** the association with coronary heart disease, myocardial infarction and major depressive disorder (*n* = 184,305; 171,875; 500,199 respectively) is shown. In **c** the association with responses to the PHQ-9 from the UK Biobank are shown, which are stratified by sex. PHQ-9 GWAS sample sizes; male 51,605-51,908; female 65,572–66,015; combined 117,177–117,907. See Supplementary Data 17b for detailed breakdown of sample sizes. PHQ9 Patient Health Questionnaire 9, SHBG Sex Hormone Binding Globulin.

fetoplacental PGS as genetic instruments and the inverse variance weighted (IVW) method to estimate effects on both coronary heart disease and myocardial infarction[38]. Mendelian randomization showed a protective effect on both outcomes (coronary heart disease, *p* = 0.002, *β* = −0.009; myocardial infarction, *p* = 0.003, *β* = −0.009;

Fig. 2b and Supplementary Data 13). The effect on myocardial infarction was confirmed with the weighted median method (*p* = 0.01, *β* = −0.01), while coronary heart disease fell on the threshold for significance using this method (*p* = 0.05, *β* = −0.008) (Supplementary Figs. 4a and 5a; Supplementary Data 14). In supplementary analyses we

found no evidence of instrument heterogeneity or horizontal pleiotropy using the Cochrane's Q test, leave one out analysis, single SNP analysis and the MR egger intercept (Supplementary Data 15 and 16 and Supplementary Figs. 4b, c and 5b, c). This protective effect is in contrast to the positive direction of effect we observed for cardiometabolic risk factors in the pheWAS analyses and underlines the importance of using orthogonal methods, such as mendelian randomization, which are more robust to environmental confounding.

Similar analysis for MDD also suggested a protective effect of the eQTLs, but was marginally outside the threshold for statistical significance using both the IVW ($p = 0.08$, $\beta = -0.003$) and weighted median ($p = 0.08$, $\beta = -0.003$) methods (Fig. 2b and Supplementary Fig 6; Supplementary Data 13 and 14). We reasoned this may either be due to sex-dependent or symptom-specific effects. To address these questions we ran Mendelian randomization analyses using sex-specific GWAS summary statistics from the patient health questionnaire 9 (PHQ-9), answered as part of the UK Biobank[39,40]. The PHQ-9 is a self-report questionnaire based on the 9 DSM criteria for a diagnosis of depression (Supplementary Data 17a). In this analysis, we identified a robust protective effect for suicidality in females ($p = 0.003$, FDR = 0.02, $\beta = -0.001$; Fig. 2c and Supplementary Data 13). This finding was confirmed with the weighted median method ($p = 0.04$, $\beta = -0.001$; Supplementary Data 14), with no evidence of heterogeneity or horizontal pleiotropy (Supplementary Fig 7; Supplementary Data 15 and 16). We also identified a significant effect in the full sample for suicidality using the IVW method, but this could not be confirmed by the weighted median method (Supplementary Fig. 8 and Supplementary Data 13 and 14).

These results suggest baseline expression of the cyan module in the placenta is protective against adult cardiovascular disease in a male/female combined sample and against suicidal ideation in females. These results are in line with a large body of literature describing a positive correlation between cardiovascular and depressive risk[41,42].

### Enrichment of differentially expressed genes and drug targets within the cyan module

Interestingly, previous work in populations exposed to instances of pathogenic inflammation, such as preterm birth or general prenatal infection, have shown an *increased* risk for both cardiovascular disease and depressive outcomes[21,22,43]. This is in contrast to our findings, where we see *protective* effects of the cyan module on these outcomes. A parsimonious explanation for this is that pathological prenatal exposures may partly confer risk for adult health outcomes by disrupting the protective effects of the cyan module, and thus Hofbauer cell function, in the placenta.

Therefore, we next asked whether placental genes differentially regulated by preterm birth, infection, or other exposures were enriched within the cyan module. We mined the published literature for studies that performed differential expression analysis in the human placenta following various exposures (Supplementary Data 18) and extracted differentially expressed genes. We found a very strong enrichment of genes differentially expressed in response to intra-amniotic infection (154 genes; $p = 5.2e-15$) and preterm birth (15 genes; $p = 2.6e-05$) in the cyan module (Fig. 3a; Supplementary Data 18). We conducted a similar analysis using the human orthologues of genes differentially expressed in the mouse placenta following two models of prenatal infection (poly I:C and Listeria monocytogenes). Both gene lists were strongly enriched in the cyan module (poly I:C- 79 genes $p = 8.7e-09$; Listeria monocytogenes- 42 genes, $p = 2.7e-06$; Fig. 3b). We then utilized data from Lu-Culligan et al.[23] and created an expression score for the cyan module in Hofbauer cells from placentas with or without an active SARS CoV-2 infection. A comparison of the two groups showed the cyan module was markedly decreased in the SARS CoV-2 group ($p < 2.22e-16$;

Fig. 3c), suggesting disruption of Hofbauer cell function during prenatal infection with SARS CoV-2.

We then asked if any drugs are known to target genes of the cyan module. We used DGIdb, which compiles gene expression effects of drugs. This unbiased analysis identified the anti-inflammatory drug, aspirin, as the drug with the most targets within the cyan module (Fig. 3d).

We then used ARACNE[44] to measure the connectivity of genes targeted by aspirin, genes differentially expressed with intra-amniotic infection or preterm birth in humans and in animal models of prenatal infection (Fig. 3e and Supplementary Data 19). All groups had a higher mean connectivity than the background list (full cyan module), while those genes differentially expressed by intra-amniotic infection in humans and in the maternal immune activation mouse model had statistically significant higher connectivity than the background of the cyan module. The aspirin group, although only consisting of 8 genes, showed a nominally significant increase in connectivity compared to the full cyan module, suggesting it may be a useful modulator of cyan module expression.

Our data advocate for a mechanism by which prenatal infection and preterm birth confer risk for cardiovascular and depressive outcomes, at least partially, through disruption or loss of the homeostatic functions of the cyan module and the cell type within which it is primarily expressed, Hofbauer cells.

## Discussion

We used a multi-modal approach integrating transcriptomics, genetics and adult health outcomes across multiple cohorts and genetic ancestries to assess the effect of inflammation-related placental gene expression on adult offspring outcomes. We used distinct complementary methods and several supplementary analyses to identify a protective effect of a placental gene co-expression module, principally expressed in Hofbauer cells, on adult depressive and cardiovascular outcomes. Our results suggest disruption or loss of the homeostatic functions served by Hofbauer cells may contribute to the increased risk of adult cardiovascular and depressive outcomes in individuals exposed to prenatal infection or born preterm.

Intra-uterine infection is a primary precipitant of preterm birth[45], which in turn is associated with a marked increase in risk for adult depression and cardiovascular disease[43,46,47]. Loss of the placenta and therefore Hofbauer cell function, is an inevitable consequence of preterm birth. A model under which this premature loss of Hofbauer cell function contributes to adverse adult health outcomes in offspring is plausible. Indeed similar mechanisms have been described for several placental mechanisms, with particularly strong evidence for endocrine functions[48]. Our cord blood analysis found the strongest association between the fetoplacental PGS and MCP-1, suggesting that it (or indeed other unmeasured secreted molecules) could act to stimulate fetal development. Premature cessation of this endocrine signaling may then contribute to adverse adult health outcomes in offspring. Hofbauer cells are understudied and future comprehensive characterizations of their function throughout pregnancy would likely further inform these mechanisms and may even point to new therapeutic or preventive strategies in preterm infants.

Intra-uterine infection can also occur in pregnancies carried to term, but adult outcomes of this population is less studied. Population studies that do not discriminate between maternally confined and intra-amniotic infections have observed an increased risk for both depression and cardiovascular disease in offspring[21,22], but future work stratifying by infection type will be critical. In our study, the transcriptional response to intra-uterine infection was highly enriched within our Hofbauer gene expression module. These genes in turn were central components of the network, suggesting severe disruption in instances of intra-uterine infection. Histological studies have indeed seen changes in Hofbauer cell distribution with intra-uterine

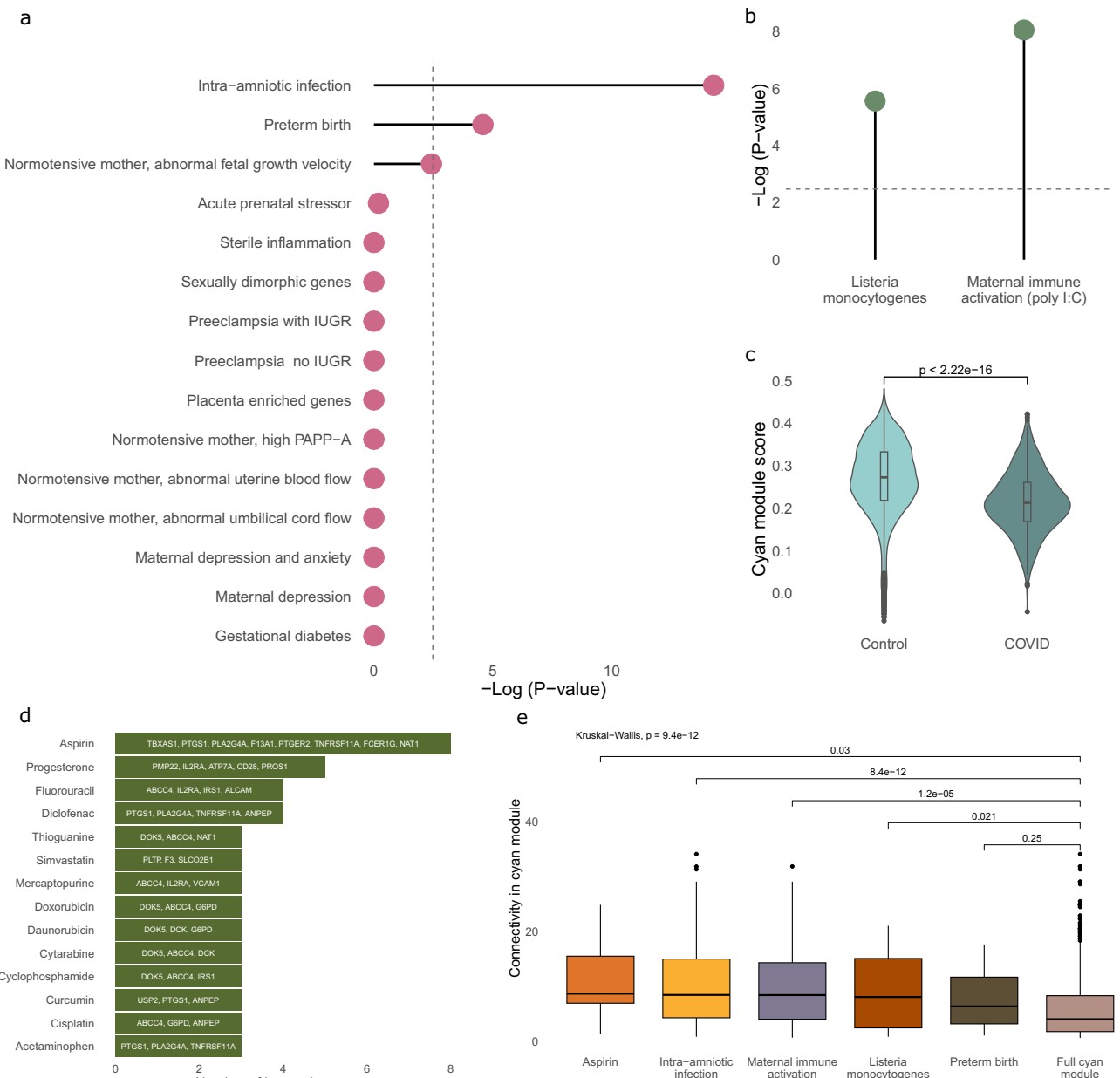

**Fig. 3 | The cyan module is highly enriched for genes differentially regulated by prenatal infection and preterm birth. a** Enrichment analysis using a Fisher's exact test for genes differentially expressed in the placenta under various conditions. Dashed line indicates the FDR-corrected threshold for multiple comparisons. **b** Enrichment analysis using a Fisher's exact test for genes differentially expressed in the mouse placenta following two models of prenatal infection, within the cyan module. Genes were converted to human orthologues prior to enrichment analysis. The dashed line indicates the FDR-corrected threshold for multiple comparisons. **c** Expression score for the cyan module within Hofbauer cells of term placental villi samples for individuals with (3513 cells) or without (40627 cells) an active SARS CoV-2 infection[23]. Groups were compared using a Wilcoxon test (two-sided). **d** Drugs annotated in the Drug-Gene Interaction Database which target members of

the cyan module. Drug names are on the y-axis with the names of genes they target in the cyan module labeled within their respective bar. **e** Boxplot for the connectivity of genes within the cyan module that are either affected by aspirin or differentially expressed under various pathogenic conditions in humans (intra-amniotic infection and preterm birth) or animal models (maternal immune activation and Listeria monocytogenes). Number of genes in each group: Aspirin (8 genes), intra-amniotic infection (152 genes), maternal immune activation (79 genes), Listeria monocytogenes (42 genes), preterm birth (15 genes), and the full cyan module (486 genes). Groups were first compared using a Kruskal–Wallis test, followed by post hoc comparisons with the full cyan module using a Wilcoxon test (two-sided). In both c and e, boxplots display, in order from bottom to top- lower extreme, lower quartile, median, upper quartile, and upper extreme.

infection[49]. Hofbauer cells also express physiologically functional Toll Like Receptors, suggesting an active role in the response to intra-amniotic infection[16]. Previous studies have even observed direct infection of Hofbauer cells by HIV, Zika and SARS-CoV-2 viruses[50–52]. Together, this suggests a promising clinical approach to reducing risk of adverse offspring health outcomes following intra-amniotic infection may be to target the cyan module and Hofbauer cells.

Of the drugs annotated in the DGIdb, the anti-inflammatory drug, aspirin, had the most targets in the cyan module. Even though these genes represented only a minority of the module's membership these genes showed a high degree of connectivity, indicating their potential to affect network integrity. Aspirin is an attractive candidate considering its breadth of use and volume of available data. In fact, aspirin is already recommended for use in pregnant women at high risk of

preeclampsia[53,54], and estimates suggest it is already used by up to 38.8% of this population in the United States[55]. It is unknown if aspirin therapeutically acts through Hofbauer cells in the context of pre-eclampsia, but a previously characterized function of Hofbauer cells is in angiogenesis. Disruptions in placental angiogenesis have also been observed with prenatal infection[56] and MCP-1, which we found to have the strongest association with the fetoplacental PGS in cord blood, is also an angiogenic regulator[57,58]. A credible hypothesis based on our results is that prenatal infection elicits a response from Hofbauer cells, which perturbs placental angiogenesis with long-term consequences for the fetus. The effects of aspirin on the cyan module, Hofbauer cell function, angiogenesis, and ultimately whether it is beneficial for pregnancies at risk of intra-uterine infection, are key future questions.

We found a female-specific effect of placental eQTLs for cyan module genes on the risk of suicidality. This is not surprising, as sex differences in the prevalence of depression[59] have been widely reported. Our results provide evidence that sexually dimorphic regulation of risk starts during early development. We did not identify any correlation of cyan module expression with fetal sex and previous studies in other tissues have not observed notable sex-biases in eQTLs[60]. Therefore, we have no evidence to suggest sex biases were present in our analyses at either the gene expression level or eQTL level. Processes downstream of the cyan module are then likely responsible for these sex differences, but future work is required to establish the nature of these processes.

Our study has limitations that must be considered. Our analysis was limited by the availability of placental functional genomic resources. Future studies with increased power for placental eQTL discovery are essential to increase the discovery power of studies such as ours. Furthermore, due to the nature of many large GWASes, summary statistics were often only available for males and females combined. As GWAS sample size increases, we anticipate the wider availability of sex-specific summary statistics will further expand the study of sex differences into powerful methods like Mendelian randomization. Another limitation to the current study is the limited number of diseases and diagnoses we considered. Future work armed with larger and more genetically diverse eQTL databases that examine an expanded catalog of diagnoses will be well placed to expand upon our results. Alternative approaches, such as using gene ontology terms to categorize inflammatory mediators, or investigating different cell type expression patterns are also promising strategies for future studies. Finally, we confined our RNA sequencing samples to term births to avoid pathological artifacts associated with preterm birth. Characterizing the developmental expression trajectories of inflammation-related gene expression patterns in the placenta will be an important future step.

In conclusion, we identified an inflammation-related gene expression module in the placenta that was highly enriched in Hofbauer cells, and created a fetoplacental PGS that specifically predicted expression of this module. We used this fetoplacental PGS to identify associations with traits in the cardiometabolic and mental health domains of the UK Biobank. Follow-up analyses using Mendelian randomization demonstrated genetically inferred cyan module expression was protective for cardiovascular and depressive outcomes. We finally showed that genes differentially regulated by both prenatal infection and preterm birth were highly enriched in the cyan module. These findings suggest that loss of Hofbauer cell function with preterm birth or its disruption with intra-amniotic infection may contribute to the increased risk of cardiovascular disease and depression in these offspring.

## Methods
### Cohorts
We used data from two population-based cohorts. The first was the Growing Up in Singapore Towards healthy Outcomes (GUSTO; Soh

et al., 2014) cohort, a Singapore-based longitudinal birth cohort that recruited mothers at least 18-years of age from the two largest maternity units in Singapore. Ethical approval for GUSTO was granted by the relevant institutional boards (DSRB reference D/09/021 and CIRB reference 2009/280/D) and written informed consent was received from all participating mothers. The second data source was the UK Biobank[61], a large adult population-based study in the UK. The UK Biobank is guided by an ethics advisory committee and informed consent was received from all participants. Approval for the UK Bio-bank was obtained by the Northwest Multicentre Research Ethics Committee (REC reference 11/NW/0382), the National Information Governance Board for Health and Social Care and the Community Health Index Advisory Group. Access to data used in the current study was obtained under application #41975. In all analyses performed, sex was defined genetically. Samples sizes for specific analyses are described in the relevant results or Supplementary Data.

### Placental sampling
Placental tissue was obtained as part of the GUSTO in line with the approved protocols. Placenta samples ($n = 44$; 2 subsequently excluded with hierarchical clustering) used in this study were derived from term births with tissue collected within 40 minutes of delivery. Exclusion criteria included chorioamnionitis, antenatal smoking (confirmed with plasma cotinine), maternal BMI greater than 30 kg/m$^2$, antenatal fasting glucose greater than 7 mmol/L or 2 hour oral glucose tolerance test result greater than 11.1 mmol/L, hypertensive disorders of pregnancy, birth prior to 37 weeks of gestation and a gestational age and sex-standardized birthweight percentile less than 10%. Placental biopsies were taken at random from 3 sites at the maternal-facing side, before removal of the maternal decidua to primarily retain the placental villous tissue for analysis. Sampling sites were then pooled and stored at −80C until further processing.

### RNA extraction and sequencing
Total RNA from the placental samples was extracted from samples using the phenol-chloroform method, followed by large RNA purification using the NucleoSpin miRNA kit (Machery-Nagel, Düren, Germany) as per manufacturer's instructions. RNA concentrations were determined using a Nanodrop spectrophotometer (Thermo Fisher Scientific, Waltham, MA, USA) and RNA integrity number (RIN) was measured using the Agilent 4200 TapeStation System (Santa Clara, CA, USA). Sequencing libraries were prepared from samples with a RIN > 6 at Novogene AIT Genomics (Singapore). In brief, ribosomal RNA was depleted with the Illumina Ribo-Zero Magnetic Kit for Human/Mouse/Rat (San Diego, CA, USA). Library preparation was done using the NEB Next Ultra Directional RNA Library Prep Kit (New England Biolabs, Ipswich, MA, USA), before sequencing was carried out using the Illumina HiSeq platform with a minimum depth of 50 million paired-end 150 bp reads. Sequencing quality was assessed using FastQC[62] and MultiQC[63]. Reads were aligned to the human hg19 genome and gene level counts assembled with STAR Aligner[64]. Samples had an average unique mapping rate of 93.5%.

### WGCNA
First raw gene counts were converted to RPKM (Reads Per Kilobase of transcript per Million mapped reads). Both protein and non-coding genes from all chromosomes were included in the analysis. We initially identified 53,010 genes, which were then filtered to those genes expressed in more than 90% of samples. This strategy resulted in 31,097 genes remaining. All of these genes passed further filtering thresholds with the goodSamplesGenes() function from the WGCNA package and were thus submitted to WGCNA.

Considering the possibility of erroneous blood clots included within the tissue, which may bias the sequencing data, we used hierarchical clustering to identify outliers. This process identified two

samples that showed a distinct grouping pattern compared to the other samples and were therefore removed. After which 42 samples and a total of 31,097 genes remained. Gene co-expression networks were constructed by automatic block-wise network construction and the module detection process implemented in the WGCNA package[19]. The soft thresholding power ($\beta = 14$) for adjacency calculation was chosen based on approximate scale-free topology before co-expression network construction. The topological overlap matrix (TOM) was built by calculating gene adjacencies using biweight mid-correlation with the chosen soft thresholding power, before signed weighted correlation networks were built. Modules were detected via hierarchical gene clustering on TOM-based dissimilarity and branch cutting using the top-down dynamic tree cut method. Eigengenes were calculated for each module and modules with high eigengene correlations (r > 0.85) were merged. Genes not incorporated to any module were assigned to the non-functional gray module. In total 28 functional modules were identified.

## Module preservation
$Z_{summary}$ statistics were calculated for module preservation analysis by combing module density-based statistics and intra-modular connectivity-based statistics and separability of modules based on a permutation test. As per the recommended thresholds we defined $Z_{summary} < 2$ as no evidence of preservation, $2 < Z_{summary} < 10$ as weak to moderate evidence of preservation and $Z_{summary} > 10$ as strong evidence of preservation.

We tested our modules for preservation in an independent dataset, GSE148241 (https://www.ncbi.nlm.nih.gov/geo/query/acc.cgi?acc=GSE148241)[28], which included 41 placenta samples (9 with early-onset severe preeclampsia and 32 healthy controls). Only the 32 healthy control samples were extracted for use in the analysis. We performed 100 permutations to reconstruct the networks using the same parameters to randomly permute module assignment in the test data.

## Gene ontology
Gene ontology enrichment for biological processes was carried out using the gprofiler2 v0.2.1 package with default settings for the Biological Processes category[65].

## Single cell and cell type enrichment
For single expression scores data were downloaded from GSE171381[23]. Data were integrated and normalized using Seurat v3.2.3 as previously described[66] pipeline and module score created using the AddModuleScore() function. We used the cell type assignments from the original study to chart cell specific expression patterns.

Cell type enrichment in datasets from Vento-Tormo et al.[8] and Suryawanshi et al.[9] was carried out using the PlacentaCellEnrich tool[67].

## Transcription factor enrichment analysis
Transcription factor enrichment analysis was performed using the Top Rank method in the ChEA3 package using the online interface and default settings[68].

## Genotyping and PGS generation
Genotyping in the GUSTO cohort was performed with the Infinium OmniExpress Exome array. Quality control was done separately for each genetic ancestry. SNPs with a call rate <95%, minor allele frequency <5%, that were non-autosomal or that failed the Hardy-Weinberg equilibrium (p-value of $10^{-6}$) were removed from the analysis. Variants discordant from their respective subpopulation in the 1000 Genomes Project reference panel were removed (Chinese- EAS with a threshold of 0.20; Indian- SAS with a threshold of 0.20; Malay-EAS with a threshold of 0.30). Samples were removed if they had ancestry or sex discrepancies, call rate <99% or showed evidence of cryptic relatedness. Data were then pre-phased with SHAPEIT v2.837

with family trio information, and imputation carried out using the Sanger Imputation Service with the 1000 G Phase 3 dataset as a reference, using the "with PBWT, no pre-phasing" (the Positional Burrows Wheeler Transform algorithm) pipeline. Imputed SNPs common to all genetic ancestries, which were bi-allelic, non-monomorphic and that had an INFO score > 0.8 were retained for downstream analysis. PGS were generated in the GUSTO cohort for all available offspring.

UK Biobank genotyping and quality control processes are comprehensively described in Bycroft et al. 2018[61]. Participants were excluded from the analysis if consent was withdrawn, genotyping data was unavailable, a genetic kinship to other participants >0.044 identified, inconsistent genetic and reported sex, or if the subject was an outlier for heterozygosity. We then identified a single participant from each genetic kinship group (genetic relatedness <0.025), based on their genomic relationship matrix (calculated using Genome-wide Complex Trait Analysis GCTA 1.93.2), which were returned to the analysis. PGS were generated in the UK Biobank for the full available sample.

## PGS generation
The fetoplacental PGS and the negative control PGS for Fig. 1d were generated as previously described[31]. In brief, SNPs located on genes in a relevant list were identified using the biomaRt package[69]. SNPs common with placental eQTLs identified by Peng et al., 2017 were retained. These eQTLs were subjected to linkage disequilibrium clumping ($r^2 < 0.2$) in GUSTO and the UK Biobank, leaving only independent loci. For PGS calculation, the number of effect alleles at a particular locus was weighted based on the effect size on gene expression identified by Peng et al., before summation within individuals to generate the relevant PGS.

Negative control PGS were also generated to assess the specificity of the fetoplacental PGS to predict cyan module expression. These negative control PGS consisted of the three WGCNA modules closest in size to the cyan module (Midnightblue (442 genes), Lightcyan (427 genes) and Salmon (607 genes)) and two random selections of 486 genes from the sequencing dataset. These negative control PGS were built in an identical fashion to the fetoplacental PGS, using placental eQTLs. As an additional control we used the cyan module to generate another PGS but weighted it based on eQTLs from the fetal cerebral cortex.

PGS for cardiovascular disease (CVD) and major depressive disorder (MDD; as provided by the psychiatric genomics consortium) was performed using GWAS summary statistics from European ancestry populations by Nikpay et al. 2015 and Howard et al. using PRSice software v2.2.11.b[70] at 10 p-value thresholds (0.00000001, 0.0000001, 0.000001, 0.00001, 0.0001, 0.001, 0.01, 0.1, 0.2 and 0.5).

## Single sample gene set enrichment analysis
Single sample gene set enrichment analysis (ssGSEA) was implemented through the GSVA (Gene Set Variation Analysis) package as previously described[71]. Scaled RPKM gene counts were used in the analysis. The effect of the fetoplacental PGS on ssGSEA score in the GUSTO RNA-seq samples was determined through multiple regression using sex and the first 3 genetic principal components as covariates in the analysis.

## Cord blood analysis
Molecular characterization of cord blood from the GUSTO cohort (sample size of between 194-251 depending on specific molecule analyzed) was conducted in duplicate using commercially available assays. Samples were randomized across plates and internal controls were used to estimate cross-plate variation. Assays with a coefficient of variation exceeding 20% across internal standards were excluded. Molecular profiles were analyzed using 1 of 3 methods: single molecule array (SIMOA; IL6, IL10, TNFa, IFN gamma, IL4), DropArray (MCP1,

TSH, insulin, VEGFA, LH, IgE, FSH, glucagon, IP10, Leptin, MIP1a, CRP, C-peptide, IL1RA, IGFBP7, prolactin, MIP1b, growth hormone, IGFBP3, IL12p40, GLP1) and enzyme-linked immunosorbent assay (ELISA; adiponectin, free testosterone, testosterone). Supplementary Data 9 describes the individual assays. SIMOA measurements were made using the SIMOA HD-1 Analyzer (Quanterix). DropArray measurements were made using the FlexMAP3D bead-based multiplex system (Luminex). Normalization was carried out across plates using a median centering method. Data with readings outside of the assay limits as indicated by the manufacturer were discarded.

## UK Biobank analysis

All analyses were restricted to unrelated individuals with self-reported sex matching genetically identified sex. Individuals with high genetic missing rates or heterozygosity were also excluded from the analysis.

We used the PHESANT package to explore associations with the fetoplacental PGS using a pheWAS framework in the UK Biobank, as described previously[33]. For instances where an outcome was measured multiple times, the variable with highest sample size was retained.

## Mendelian randomization

Mendelian randomization was conducted as previously described using the TwoSampleMR package v0.5.6[36,37]. In all MR analyses placental eQTLs were used as the exposure. Outcome SNPs from MDD[72], coronary heart disease[38] and myocardial infarction[38] GWAS were obtained through the IEU GWAS database. The MDD summary statistics without 23 and me subjects were used in the analysis. Sex-specific GWAS summary statistics for the PHQ-9 in European subjects were downloaded from the Neale lab website[40]. Further information on this data can be found in Supplementary Data 17a and 17b. Exposure and outcome data were harmonized to the same effect allele, with ambiguous SNPs removed from the analysis. We then carried out a fixed effects meta-analysis of the genetic instruments using the IVW (inverse variance weighted) method. These results were then confirmed using the more conservative weighted median method which assumes at least 50% of the instruments used are valid. Heterogeneity and horizontal pleiotropy were then assessed using several methods including the Cochran Q statistic, leave one out analysis, single SNP analysis and the MR egger intercept.

## MAGMA GWAS enrichment

We conducted GWAS enrichment using MAGMA v1.10[73] using a window of ±50 Kb around each gene. We used SNP annotations from the 1000 genomes European dataset and gene annotations from the NCBI website build 38.

## Drug interactions

Drug-gene interactions were interrogated using the online interface of the Drug-Gene Interaction Database v4.2.0[74].

## Enrichment analysis

Enrichment for differentially expressed genes in the cyan module was done using a Fisher's exact test as implemented with the GeneOverlap (v1.30.0) package. Genes differentially expressed in the mouse placenta were first converted to human orthologues using the gprofiler online interface[65].

## Network connectivity

To investigate the connectivity of the genes in a particular module we used ARACNE (Algorithm for the Reconstruction of Accurate Cellular Networks[44]) to identify significant gene by gene associations within the modules based on their mutual information. For each ARACNE-derived unweighted network, the connectivity scores of the hub genes were computed.

## Statistical analysis

All analysis unless otherwise stated were carried out using R v4.1.1 and Rstudio v1.4.1717. Regression analyses were used to analyze the association between the fetoplacental PGS and cord blood molecules (GUSTO), ssGSEA (GUSTO) and UK Biobank outcomes using lm() or glm() functions. Specific covariates for each analysis are described in the dedicated figure legends. The cord blood measurements were log transformed before inclusion in a linear regression analysis with the fetoplacental PGS as the predictor. In all relevant genetic regression analyses population stratification was accounted for by including genetic principal components as covariates. A significance threshold of 0.05 was used throughout, with the Benjamini-Hochberg method (False Discovery Rate; FDR) used to correct for multiple comparisons.

## Reporting summary

Further information on research design is available in the Nature Portfolio Reporting Summary linked to this article.

## Data availability

The data in this manuscript from the UK Biobank and GUSTO cohorts are available dependent on successful application the respective data access committees. Data for WGCNA validation were obtained through GEO accession number GSE148241[28]. GWAS summary statistics used in this study were obtained from publicly available repositories. UK Biobank PHQ-9 summary statistics were obtained through the Neale lab website http://www.nealelab.is/uk-biobank/. The depression summary and cardiovascular-related summary statistics were obtained through the MRC IEU database under accession numbers ieu-b-102, ieu-a-7, and ieu-a-798, respectively. Single-cell RNA-seq data for cell type enrichment were obtain through GSE182381, GSE171381 or at https://placentacellenrich.gdcb.iastate.edu/. Datasets used in PRS generation include the Hg19 build obtained through https://www.ncbi.nlm.nih.gov/datasets/genome/GCF_000001405.13/ and the 1000 genomes dataset via https://www.internationalgenome.org/data. All other data generated or used in this study are provided in the supplementary material.

## Code availability

Code for these analyses was in line with vignettes for all packages mentioned in the methods. Code to run the pheWAS analysis can be found at https://github.com/MRCIEU/PHESANT. Code to run the Mendelian randomization analysis can be found at https://mrcieu.github.io/TwoSampleMR/.

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

## Acknowledgements

We would like to acknowledge both the GUSTO and UK Biobank investigators, staff, researchers and in particular the participants. This research has been conducted using the UK Biobank Resource under Application Number 41975. Funding for this study was provided through funding of the Depression Task Force of the Hope for Depression Research Foundation to MJM.

## Author contributions

E.F. planned the study, performed the analysis, interpreted the data, and wrote the initial draft of the manuscript, with editing from M.J.M. who provided funding. H.C. provided feedback on the manuscript. M.J.S. performed WGCNA. H.E.J.Y. and P.L. coordinated and prepared placental samples for RNA-seq. S.Y.C. oversaw the sample selection for RNA-seq. Placentas were collected in accordance with protocols developed by J.C. RNA-seq data were analyzed by E.F., N.O.T., and M.J.M. Calculation of PGS was performed by Z.W., S.P., and I.P.; Y.S.C., P.D.G., and M.J.M. designed the GUSTO cohort study and obtained funding. All authors approved the final version of the manuscript.

## Competing interests

The authors declare no competing interests.
