## [Peer Review File · Nature Communications]

Hofbauer cell function in the term placenta associates with adult cardiovascular and depressive outcomesREVIEWER COMMENTS

Reviewer #1 (Remarks to the Author):

Reviewer Overall Comments: The goal of this publication is to generate an expression signature of homeostatic inflammation of the placenta, and to identify genetic variants associated with this inflammation to create a polygenetic risk score (PGS) of placental inflammation. This score is used to demonstrate that genetic variation related to placental inflammation is linked to later life health outcomes including major depressive disorder and cardiovascular disease. This is a very novel application of polygenic risk scores to evaluate the DOHAD hypothesis, and is particularly significant because it provides a path to study later life health outcomes, which is not feasible using more traditional approaches. I have some concerns about how the data presented in the study supports the conclusions presented by the authors. My major critique of the manuscript lies within the authors presentation of "homeostatic inflammation", which the authors claim they quantify through one of the WGCNA modules (Cyan module). The phrase "homeostatic inflammation" itself is not well defined or precise in the context of placental biology, and their cyan gene expression module does not seem to be a strong marker of inflammation based on the GO findings. Their work does support their conclusion that the cyan gene module is predicted by their curated PGS. think there could overall be more precision in how this data is presented and some clarification as to how specific modules, pathways, diseases excetera were selected for discussion and for this narrative, as they do not appear to be the top hits.

1. The introduction of this manuscript highlights the importance of the placenta in shaping lifelong health, then discusses how prenatal infection drives adult behavioral outcomes. The authors provide an example of important immune cells in the brain and highlight an important placental immune cell (Hofbauer cells). However, the introduction does not provide much detail about the important role of placental inflammation in pregnancy maintenance, or how this inflammation changes across gestation. Briefly, the first and third trimesters of pregnancy are thought of as pro-inflammatory stages, whereas the 2nd trimester represents an anti-inflammatory stage. Some baseline inflammation is essential for placental angiogenesis in the first trimester, and for appropriate initiation of labor. Please consider including some additional background on this nuanced topic (<https://doi.org/10.1530/REP-16-0453>; doi: 10.1111/j.1749-6632.2010.05938.x; <https://doi.org/10.1038/s41574-020-0372-6>)

2. Of the 28 gene modules identified through WGCNA, the authors only discuss the cyan module in this paper. Other papers that utilize WGCNA traditionally present the significant findings across all modules. I am wondering if any of the other markers were enriched for the same inflammation pathways that the authors highlight in Figure 1B. (especially since these pathways are so large). The Cyan module itself was associated with a total of 94 different pathways (Table 3) and a subset of these pathways related to immune signaling are presented in Figure 1B, which the authors use to justify the use of this module as a marker of homeostatic inflammation. However, the pathways presented in this figure seem to be cherry picked, and do not reflect the top pathways most significantly enriched within the module, which are "integral/intrinsic component of the membrane, plasma membrane, and cell periphery". These are GO Cellular component terms, which conflicts with the authors methods section saying they did enrichment of GO:BP terms. The top GO:BP terms are response o a stimulus and regulation of multicellular organismal process. Further explanation of why the cyan module was a unique signature of inflammation is warranted. How were these pathways selected and this defined?

3. The authors highlight several inflammatory genes within the cyan module including interleukin and TNF genes. Can the authors explain the rationale of using this data driven approach to identify this broad list of genes that does contain some inflammatory genes, vs. starting off with a an a priori list of homeostatic inflammatory markers based on the literature or a specific GO pathway, and then performing subsequent downstream analyses (including generating the inflammatory PGS and then the PHEWAS study) using those genes?

4. The authors have generated gene expression signatures of homeostatic inflammation using term, non pathological samples. This is a major strength of their analysis because there is a significant research need to understand how subtle shifts in the in-utero environment may influence lifelong health. The authors appropriately describe their exclusion criteria including smoking, high BMI, maternal hypertension or signs of GDM, and low for gestational age infants. The authors present key covariate data including maternal ethnicity, maternal education and sex in Table 1. However, they do not mention if they excluded participants presenting with acute or chronic chorioamnionitis, which would be a source of inflammatory signatures within their dataset. Can the authors comment if this information is available for participants and can be included?

5. The authors present information about delivery method (all vaginal) in their table. However, there is no description of labor status (induced or augmented labor vs or normal labor). Labor status and delivery method are associated with massive changes to the placental transcriptome in other analyses and could be a major confounder. It would be important for the authors to (A) present data about the labor status of these individuals and (B) to see if the WGCNA modules (especially cyan) were associated with labor status in this cohort (include in Figure 1E).

6. Processes describing placental sampling and RNA isolation and sequencing are adequately described within the methods. The authors do not present details on filtering, and the number of genes included in the WGCNA (~31K genes) seems high. How was filtering applied within their gene expression data? Were only protein coding genes used? The creators of WGCNA highlight the importance of gene filtering in their tutorials/documentation, and suggest that only 10-15K genes should be expressed within a given tissue and included in WGCNA (<https://peterlangfelder.com/2018/11/25/filtering-and-collapsing-data/>)

7. The authors define Placental cell types using 2 papers (Vento Tormo et al, 2018, and Suyawanshi et al, 2018). Both of these manuscripts develop cell signatures using first trimester placental samples. Other studies have shown distinct differences in placental gene expression, and methylation of genes between trimesters, as well as cellular proportions (<https://doi.org/10.1186/s12864-020-07186-6>; doi:10.1371/journal.pone.0033294). Of particular note, Yuan et al identified distinct difference in Hofbauer cells between first trimester and term placentas. This is of concern given all the placental samples in this dataset are from term placentas, so the timing doesn't match. Do the authors think this could influence their cell type enrichment results?

8. The authors performed a PHEWAS to identify how the PGS was associated with later life health outcomes, with complete results presented in figure 2A and table 9. Their conclusions from this analysis are that there are an enrichment of significant phewas traits associated with cardiometabolic and mental health outcomes in the UK Biobank data. However, the results presented in Figure 2 show that these traits are most strongly associated with physical measures, including the top hits including hip circumference and trunk fat percentage. Can the authors better explain how cardiometabolic and mental outcomes were selected based on these results? Was this from an a priori hypothesis? If so, why not just test those specific domains instead of performing a full PHEWAS?

9. The authors highlight that they found a strong signal and enrichment of DEGs that were previously associated with intra-amniotic infection (154 genes; 395 $p=5.2e-15$) and preterm birth (15 genes; $p=2.6e-05$) in the cyan module (Figure 3A; Table 20). These DEGs are defined from Pereyra et al, 2019 and Motomura et al, 2021. These studies were conducted within the chorioamniotic membranes and not the placenta itself. It would be beneficial to go through the studies presented in table 20 and clarify what tissue the data was generated within, and to reflect this tissue in the presentation of their findings.

Reviewer #2 (Remarks to the Author):

This is a very interesting manuscript exploring the relationship between homeostatic inflammation in the placenta and risk for cardiovascular and depression outcomes. Overall, the paper is well written and addresses an important problem; this study is important and will likely make an impact for much future work.

Major

- a little more details in the methods are needed. I will expand on each of them below.
- For example, are the placental samples (line 106) from GUSTO? Or some other IRB approved study?
- Was the total RNA extracted from placenta, blood, both? Line 116
- What was the issue with the 2 samples dropped - line 132
- In the PGS generation, which individuals were used? Was this women only? Everyone? Lines 175-197 need a little more detail.
- The gene list used for PGS generation should be provided as a supplemental table. It is not reproducible without it.
- What is the final SNP list? Same SNPs used in GUSTO and UKBB? Lines 189-195
- How does the PGS for CVD and MDD overlap with the placental list?
- Which PGS is being referred to in line 220? Placental? CVD? MDD? This was repeatedly confusing. I think that the authors should consider using the type of PGS as a subscript in the name (PGS_CVD for example).
- What cytokines were generated (line 250). Detail is very limited here.
- How do you measure homeostatic inflammation (line 258-9)?
- How did the authors decide on making 3 PGS (lines 296-301)? It seems very limited.
- Why is there a range (line 309)? I'm not sure I understand what this means.
- How is the association between the fetoplacental PGS and diagnoses different from PheWAS? (line 329)
- The risk between inflammation and CVD is confusing. Sometimes, the paper says increased risk and sometimes protective. I think this needs to be clarified throughout.
- Which effects are sex dependent (line 461)?
- What is the ancestry of the summary statistics used for the generation of PGS? Does it match the study samples here?

Minor

- Methods, line 98, consider adding a reference for UK Biobank
- Why is 23andme excluded? This means that you may need to provide the sumstats files if they are not available. Otherwise, the results are not reproducible.
- Why is Table 8 the first table mentioned (line 211). Typically, the tables are numbered in the order they appear in the paper.
- I believe GSEA is a typo on line 201 where it says GSVA
- When referring to sex of the samples, I assume these are the offspring (line 261). But it does not say that. It must be since only women give birth, but it is these types of details that I think need to be added.

Reviewer #1 (Remarks to the Author):

Reviewer Overall Comments: The goal of this publication is to generate an expression signature of homeostatic inflammation of the placenta, and to identify genetic variants associated with this inflammation to create a polygenetic risk score (PGS) of placental inflammation. This score is used to demonstrate that genetic variation related to placental inflammation is linked to later life health outcomes including major depressive disorder and cardiovascular disease. This is a very novel application of polygenic risk scores to evaluate the DOHAD hypothesis, and is particularly significant because it provides a path to study later life health outcomes, which is not feasible using more traditional approaches. I have some concerns about how the data presented in the study supports the conclusions presented by the authors. My major critique of the manuscript lies within the authors presentation of "homeostatic inflammation", which the authors claim they quantify through one of the WGCNA modules (Cyan module). The phrase "homeostatic inflammation" itself is not well defined or precise in the context of placental biology, and their cyan gene expression module does not seem to be a strong marker of inflammation based on the GO findings. Their work does support their conclusion that the cyan gene module is predicted by their curated PGS. think there could overall be more precision in how this data is presented and some clarification as to how specific modules, pathways, diseases excetera were selected for discussion and for this narrative, as they do not appear to be the top hits.

We thank the Reviewer for their time and effort spent evaluating our manuscript. We sincerely appreciate the positive and constructive comments, which we believe have greatly improved our study. Please find a point-by-point response to each of the comments below.

1. The introduction of this manuscript highlights the importance of the placenta in shaping lifelong health, then discusses how prenatal infection drives adult behavioral outcomes. The authors provide an example of important immune cells in the brain and highlight an important placental immune cell (Hofbauer cells). However, the introduction does not provide much detail about the important role of placental inflammation in pregnancy maintenance, or how this inflammation changes across gestation. Briefly, the first and third trimesters of pregnancy are thought of as pro-inflammatory stages, whereas the 2nd trimester represents an anti-inflammatory stage. Some baseline inflammation is essential for placental angiogenesis in the first trimester, and for appropriate initiation of labor. Please consider including some additional background on this nuanced topic (<https://doi.org/10.1530/REP-16-0453>; doi: 10.1111/j.1749-6632.2010.05938.x; <https://doi.org/10.1038/s41574-020-0372-6>)

We thank the Reviewer for this helpful suggestion, we have added several lines to the revised introduction describing the importance of inflammatory mediators in typical placental development and have referenced the articles suggested by the Reviewer.

Line 55-59:

“Indeed inflammation is a critical regulator of placental and fetal development, with the first trimester often conceptualized as pro-inflammatory and later periods as anti-inflammatory¹⁰⁻¹². Inflammatory mediators are also critical for triggering uterine contractions and subsequent fetal delivery^{11,12}. Furthermore, loss of immune cells in animal models is sufficient for pregnancy termination¹¹.”

2. Of the 28 gene modules identified through WGCNA, the authors only discuss the cyan module in this paper. Other papers that utilize WGCNA traditionally present the significant findings across all modules. I am wondering if any of the other markers were enriched for the same inflammation pathways that the authors highlight in Figure 1B. (especially since these pathways are so large). The Cyan module itself was associated with a total of 94 different pathways (Table 3) and a subset of these pathways related to immune signaling are presented in Figure 1B, which the authors use to justify the use of this module as a marker of homeostatic inflammation. However, the pathways presented in this figure seem to be cherry picked, and do not reflect the top pathways most significantly enriched within the module, which are “integral/intrinsic component of the membrane, plasma membrane, and cell periphery”. These are GO Cellular component terms, which conflicts with the authors methods section saying they did enrichment of GO:BP terms. The top GO:BP terms are response o a stimulus and regulation of multicellular organismal process. Further explanation of why the cyan module was a unique signature of inflammation is warranted. How were these pathways selected and this defined?

The Reviewer raises important points. We used the term “homeostatic inflammation” in an attempt to underscore the importance of inflammation in typical development. We now realize that in our attempt to do this, we lost the subtlety and nuance required for the presentation of such results. In light of the Reviewer’s comments we have directed the narrative of the study away from “homeostatic inflammation” and toward “inflammation-related gene expression patterns”, “Hofbauer cell function” or simply “the cyan module”, where appropriate. We have also taken several steps in the revised manuscript to clearly characterize all WGCNA modules, our motivation for choosing the cyan module and what the cyan module represents. We note the specific changes to the manuscript below:

1. We shift the narrative from ‘homeostatic inflammation’ to ‘Hofbauer cell function’. This includes changes to text throughout the manuscript and title.
 - New manuscript title: “Hofbauer cell function in the term placenta is protective against adult cardiovascular and depressive outcomes”
2. We now provide the top 10 enriched GO terms for each WGCNA module. We used GO terms under the Biological Processes category as they were the most

functionally informative with respect to general inflammatory processes. We provide this information in Supplementary table 3 and list the enriched terms for the cyan module below in Reviewer table 1.

Module	P-value	GO term ID	GO term
cyan	7.45E-05	GO:0002376	Immune system process
cyan	9.04E-05	GO:0050896	Response to stimulus
cyan	0.000131	GO:0002682	Regulation of immune system process
cyan	0.000795	GO:0051239	Regulation of multicellular organismal process
cyan	0.001821	GO:0032501	Multicellular organismal process
cyan	0.003449	GO:0042110	T cell activation
cyan	0.00417	GO:0001775	Cell activation
cyan	0.009341	GO:0050865	Regulation of cell activation
cyan	0.011695	GO:0045321	Leukocyte activation
cyan	0.013125	GO:0001817	Regulation of cytokine production

Reviewer table 1 GO enriched terms for the cyan module

3. We used these GO results to identify modules that were broadly associated with inflammatory processes. This approach identified five modules, for which GO terms we present in Supplementary table 3.
4. We then characterized the cell type expression of these five inflammation-related modules in single cell RNA-seq data from term placental villus tissue. We present these data as a heatmap in Figure 1C of the revised manuscript and below in Reviewer figure 1. This data shows that of the 5 inflammation related module, the cyan module has the strongest cell-type specificity.
5. On the basis of these analyses and in response to the Reviewer's concern we have revised our explanation for our choice of the cyan module for further study in lines 104-113:
 - "We first used gene ontology analysis to functional classify all modules, which identified five modules enriched for terms related to inflammation (**Supplementary Table 3**). We then built expression scores for each of the inflammation-related modules in a scRNA-seq dataset of term control placental villous tissue ¹⁹ (**Figure 1B**). The cyan module had the strongest cell type specificity and was primarily expressed in Hofbauer cells, a placental macrophage localized to the villous ¹⁴ with important homeostatic ^{14,20-22} and pathogenic roles ^{14,23}. The cyan module was the largest inflammation-related module (486 genes) and also had the strongest preservation in an independent dataset ²⁴ (**Supplementary Fig 1C**). Together with the increasingly appreciated role of tissue-specific macrophages in homeostasis ²⁵, these characteristics made the cyan module the most attractive candidate for further study using a functional genomics approach."

Reviewer figure 1 Single cell expression of the five inflammation-related modules in term placental villus scRNA-seq data

3. The authors highlight several inflammatory genes within the cyan module including interleukin and TNF genes. Can the authors explain the rationale of using this data driven approach to identify this broad list of genes that does contain some inflammatory genes, vs. starting off with a an a priori list of homeostatic inflammatory markers based on the literature or a specific GO pathway, and then performing subsequent downstream analyses (including generating the inflammatory PGS and then the PHEWAS study) using those genes?

The Reviewer raises an interesting idea. Indeed, in our early iterations of the design for this study we did consider using GO terms to define inflammation-related genes for our

analysis. We moved away from this approach and toward the data driven WGCNA approach for several reasons.

First there are many GO terms related to inflammation and we struggled to find a solid basis to choose one. We considered running many analyses using different GO terms, but this would have hugely inflated our number of comparisons and the multiple correction burden. Second WGCNA has been shown in many studies to identify modules of genes enriched for disease processes without *a priori* assumptions. Third most GO terms are primarily annotated using data from adult tissues and we were unsure how well they would capture inflammation related processes in the placenta. Finally we were particularly keen to study cell-specific expression patterns and reasoned these would be better constructed using WGCNA.

However, we do agree with the Reviewer- using a GO term or another approach may well be useful. We have now suggested this as possible future work in the discussion.

Line 318-319:

"Alternative approaches, such as using gene ontology terms to categorize inflammatory mediators, or investigating different cell type expression patterns are also promising strategies for future studies."

4. The authors have generated gene expression signatures of homeostatic inflammation using term, non pathological samples. This is a major strength of their analysis because there is a significant research need to understand how subtle shifts in the in-utero environment may influence lifelong health. The authors appropriately describe their exclusion criteria including smoking, high BMI, maternal hypertension or signs of GDM, and low for gestational age infants. The authors present key covariate data including maternal ethnicity, maternal education and sex in Table 1. However, they do not mention if they excluded participants presenting with acute or chronic chorioamnionitis, which would be a source of inflammatory signatures within their dataset. Can the authors comment if this information is available for participants and can be included?

We thank the Reviewer for pointing this out, indeed chorioamnionitis was an exclusion criteria for the RNA-seq analysis and we have now added this to the methods section.

Line 354-358:

"Exclusion criteria included chorioamnionitis, antenatal smoking (confirmed with plasma cotinine Ng et al., 2019), maternal BMI greater than 30kg/m², antenatal fasting glucose greater than 7 mmol/L or 2 hour oral glucose tolerance test result greater than 11.1 mmol/L, hypertensive disorders of pregnancy, birth prior to 37 weeks of gestation and a gestational age and sex-standardized birthweight percentile less than 10%."

5. The authors present information about delivery method (all vaginal) in their table. However, there is no description of labor status (induced or augmented labor vs or normal labor). Labor status and delivery method are associated with massive changes to the placental transcriptome in other analyses and could be a major confounder. It would be important for the authors to (A) present data about the labor status of these individuals and (B) to see if the WGCNA modules (especially cyan) were associated with labor status in this cohort (include in Figure 1E).

We agree with the Reviewer, labor status is an important factor and have now included it in the table of sample characteristics (Supplementary table 1). From our placental sampling, 57.5% of women underwent spontaneous labor and this was not correlated with the fetoplacental PGS. As suggested by the Reviewer we have included labor status in the correlation plot with the fetoplacental PGS as seen in Supplementary figure 1E and Reviewer figure 2. We would also like to clarify that only 72% of the births were vaginal (Supplementary table 1), but delivery mode was also not correlated with the fetoplacental PGS (Supplementary figure 1E and Reviewer figure 2).

Reviewer figure 2 Correlations between environmental variables and the fetoplacental PGS

6. Processes describing placental sampling and RNA isolation and sequencing are adequately described within the methods. The authors do not present details on filtering, and the number of genes included in the WGCNA (~31K genes) seems high. How was filtering applied within their gene expression data? Were only protein coding genes used? The creators of WGCNA highlight the importance of gene filtering in their tutorials/documentation, and suggest that only 10-15K genes should be expressed within a given tissue and included in WGCNA

(<https://peterlangfelder.com/2018/11/25/filtering-and-collapsing-data/>)

We thank the Reviewer for pointing this out and have now added detailed information about gene filtering to the revised manuscript. We also note the majority of our modules, including the cyan module, were preserved in an independent dataset (Supplementary figure 1C).

Line 378-382:

“Both protein and non-coding genes from all chromosomes were included in the analysis. We initially identified 53,010 genes, which were then filtered to those genes expressed in more than 90% of samples. This strategy resulted in 31,097 genes remaining. All of these genes passed further filtering thresholds with the `goodSamplesGenes()` function from the WGCNA package and were thus submitted to WGCNA.”

7. The authors define Placental cell types using 2 papers (Vento Tormo et al, 2018, and Suyawanshi et al, 2018). Both of these manuscripts develop cell signatures using first trimester placental samples. Other studies have shown distinct differences in placental gene expression, and methylation of genes between trimesters, as well as cellular proportions (<https://doi.org/10.1186/s12864-020-07186-6>; doi:10.1371/journal.pone.0033294). Of particular note, Yuan et al identified distinct difference in Hofbauer cells between first trimester and term placentas. This is of concern given all the placental samples in this dataset are from term placentas, so the timing doesn't match. Do the authors think this could influence their cell type enrichment results?

We acknowledge the importance of the Reviewer's comment and have accordingly revised our manuscript. As mentioned in our response to a previous comment, we have rerun these analyses using scRNA-seq data from term placental villous samples from Lu-Culligan *et al* 2021⁶ and find the cyan module is still specifically expressed in Hofbauer cells. We present this data in Figure 1B of the revised manuscript and above in Reviewer figure 1.

8. The authors performed a PHEWAS to identify how the PGS was associated with later life health outcomes, with complete results presented in figure 2A and table 9. Their conclusions from this analysis are that there are an enrichment of significant phewas traits associated with cardiometabolic and mental health outcomes in the UK Biobank data. However, the results presented in Figure 2 show that these traits are most strongly associated with physical measures, including the top hits including hip circumference and trunk fat percentage. Can the authors better explain how cardiometabolic and mental outcomes were selected based on these results? Was this from an a priori hypothesis? If so, why not just test those specific domains instead of performing a full PHEWAS?

We appreciate the opportunity to clarify this important point.

We were struck by the enrichment of physical measures (e.g. hip circumference, trunk fat percent) and mental health measures (e.g. nervous feelings) among the significant pheWAS results. There is a well-described relation between these traits and cardiovascular disease ⁷ or mental health disorders ⁸. We considered these results exciting for two primary reasons. First the foundations of the DOHaD hypothesis were built on associations with early life events, such as a low birth weight and adult cardiovascular outcomes, with later work implicating early life events in risk of mental health disorders. Second, prenatal infection has also been associated with an increased risk of later life cardiovascular and mental health disorders. Together this prompted us to look at cardiovascular and mental health diagnoses.

We have clarified these points in the revised manuscript:

Line 168-173:

“We noted several of the pheWAS significant associations were risk factors for cardiometabolic (e.g. “hip circumference” and “trunk fat percent” ³⁰) or mental health disorders (e.g. “nervous feelings” ³¹). The discovery of a relation between perinatal events and adult risk of cardiovascular disease and mental health disorders, such as major depressive disorder (MDD), has provided much of the foundation on which the DOHaD hypothesis has been built. Furthermore prenatal infection has been associated with an increased risk of both cardiovascular disorders ¹⁸ and MDD ¹⁷.”

In an effort to streamline the narrative we have removed the analysis of diagnoses in the UK Biobank. We believe the pheWAS and Mendelian randomization results are the most central to our manuscript and the diagnoses analysis served to somewhat confuse the narrative.

We acknowledge future studies more comprehensively studying diseases and disorders will be useful in delineating the effects of the placenta on long term health outcomes. We have added text to the discussion section indicating this sentiment:

Lines 315-317

“Another limitation to the current study is the limited number of diseases and diagnoses we considered. Future work armed with larger eQTL databases and larger sequencing datasets that examine an expanded catalogue of diagnoses will be important next steps.”

9. The authors highlight that they found a strong signal and enrichment of DEGs that were previously associated with intra-amniotic infection (154 genes; 395 $p=5.2e-15$) and preterm birth (15 genes; $p=2.6e-05$) in the cyan module (Figure 3A; Table 20). These DEGs are defined from Pereyra et al, 2019 and Motomura et al, 2021. These studies were conducted within the chorioamniotic membranes and not the placenta itself. It would be beneficial to go through the studies presented in table 20 and clarify what tissue the data was generated within, and to reflect this tissue in the presentation of their findings.

We agree and have now clarified the specific site these DEGs were generated from, see Supplementary table 18 of the revised manuscript and below in Reviewer table 2.

We have also conducted additional enrichment analysis using differentially expressed genes (DEGs) in the placenta of two animal models of prenatal infection (maternal immune activation with poly I:C and maternal infection with *Listeria monocytogenes*). The human orthologues of DEGs from both of these datasets were highly enriched in the cyan module ($p= 8.7e-09$ and $2.7e-06$ for poly I:C and *Listeria monocytogenes*, respectively; Reviewer figure 3a and Figure 3b of the revised manuscript). We have also added an analysis where we show a markedly decreased expression of the cyan module in Hofbauer cells from the placental villous of pregnancies with an active SARS-CoV-2 infection (Reviewer figure 3b and Figure 3c of the revised manuscript). We finally add those genes differentially expressed in the two animal models and present within the cyan module, to our connectivity analysis (Reviewer figure 3c and Figure 3e of the revised manuscript).

Human studies			
Outcome	Study	Tissue used in study	No. DEGs
Placenta enriched genes	Gong et al, 2021	Placental biopsy	71
Normotensive mother, high PAPP-A	Gong et al, 2021	Placental biopsy	67
Normotensive mother, abnormal fetal growth velocity	Gong et al, 2021	Placental biopsy	41
Normotensive mother, abnormal uterine blood flow	Gong et al, 2021	Placental biopsy	42
Normotensive mother, abnormal umbilical cord flow	Gong et al, 2021	Placental biopsy	40
Gestational diabetes	Sober et al, 2015	Middle region of the placenta	2
Preeclampsia no IUGR	Sober et al, 2015	Middle region of the placenta	119
Preeclampsia with IUGR	Sober et al, 2015	Middle region of the placenta	55
Depression and anxiety	Litzky et al, 2018	Placenta proximal to cord insertional site	108
Maternal depression	Litzky et al, 2018	Placenta proximal to cord insertional site	107
Acute prenatal stressor	Nomura et al, 2021	Placental biopsy	3983
Sexually dimorphic expressed genes	Gonzalez et al, 2018	Chorionic villi	36
Preterm birth	Pereyra et al, 2019	Chorioamniotic membrane	270
Intra-amniotic infection	Motomura et al, 2021	Chorioamniotic membrane	5384
Sterile inflammation	Motomura et al, 2021	Chorioamniotic membrane	6
Mouse studies			
Maternal immune activation (poly I:C)	Zengeler et al, 2023	Full mouse placenta at E12	2700 (after conversion to human orthologues)
Listeria monocytogenes	Connor et al, 2022	Full mouse placenta at E18.5	1331 (after conversion to human orthologues)

Reviewer table 2 Description of studies used for enrichment analyses

Reviewer figure 3 a) Enrichment of genes differentially expressed in the mouse placenta following exposure to *Listeria monocytogenes* or poly:I:C in the cyan module. b) Reduced expression of the cyan module in Hofbauer cells of term placentas during an active SARS CoV-2 infection. c) Connectivity of genes targeted by aspirin, genes differentially expressed by intra-amniotic infection (in human chorioamniotic membranes), maternal immune activation (with poly I:C in the mouse placenta), *Listeria monocytogenes* (in the mouse placenta) or in the human preterm placenta, compared to all genes within the cyan module.

We would finally once more like to thank the Reviewer for their time and thoughtful, constructive comments.

Reviewer #2 (Remarks to the Author):

This is a very interesting manuscript exploring the relationship between homeostatic inflammation in the placenta and risk for cardiovascular and depression outcomes. Overall, the paper is well written and addresses an important problem; this study is important and will likely make an impact for much future work.

We would first like to thank the Reviewer for their time and positive, constructive comments on our manuscript. We sincerely believe their comments have greatly improved the manuscript. Please find a point-by-point response to each of the comments below.

Major

- a little more details in the methods are needed. I will expand on each of them below.
- For example, are the placental samples (line 106) from GUSTO? Or some other IRB approved study?

Yes the Reviewer is correct placental samples were obtained as part of the GUSTO study, which we now clearly state in the revised manuscript.

Line 341-343

"Ethical approval for GUSTO was granted by the relevant institutional boards (DSRB reference D/09/021 and CIRB reference 2009/280/D) and written informed consent was received from all participating mothers."

Line 352:

"Placental tissue was obtained as part of the GUSTO in line with the approved protocols."

- Was the total RNA extracted from placenta, blood, both? Line 116

RNA was extracted from placental tissue. We have clarified this point in the revised manuscript.

Line 363:

"Total RNA from the placental samples was extracted using the phenol-chloroform method"

- What was the issue with the 2 samples dropped - line 132

In our experience outliers in this context generally result from a disproportionately large blood clot accidentally included with the tissue. The reason for exclusion of these samples was based on a hierarchical clustering analysis. We have now added these details to the methods section of the revised manuscript.

Line 383-386:

"Considering the possibility of erroneous blood clots included within the tissue, which may bias the sequencing data, we used hierarchical clustering to identify outliers. This process identified two samples that showed a distinct grouping pattern compared to the other samples and were therefore removed."

- In the PGS generation, which individuals were used? Was this women only? Everyone?
Lines 175-197 need a little more detail.

PGS were generated in all available subjects of the UK Biobank and all children in the GUSTO cohort. We have now clarified this in the methods section. We thank the Reviewer for noticing this ambiguity.

Line 434-435:

"PGS were generated in the GUSTO cohort for all available offspring."

Line 442:

"PGS were generated in the UK Biobank for the full available sample."

- The gene list used for PGS generation should be provided as a supplemental table. It is not reproducible without it.

We have now included Supplementary tables in the revised manuscript to describe the genes within the cyan module (Supplementary table 2), the SNPs used to generate the PGS in the GUSTO and UK Biobank cohorts (Supplementary table 4) and the eQTLs for these genes used for mendelian randomization (Supplementary table 5).

- What is the final SNP list? Same SNPs used in GUSTO and UKBB? Lines 189-195

The majority of SNPs were indeed included in both the GUSTO and UK Biobank with some differences due to the population structure and genotype array used. We have added the SNPs used to generate each PGS to Supplementary table 4.

- How does the PGS for CVD and MDD overlap with the placental list?

The Reviewer raises an interesting question. We added an additional analysis to address this point by using a principal component regression approach implemented through MAGMA¹¹. We annotated both the MDD and coronary artery disease (CAD) GWAS summary statistics to the gene level using a window of 50Kb around each gene. We then used principal component regression as implemented in the MAGMA package and did not observe any enrichment of the cyan module within either the MDD or CAD GWAS. We have added this analysis to the results and methods sections:

Line 503-506 (methods section):

"MAGMA GWAS enrichment

We conducted GWAS enrichment using MAGMA v1.10¹¹ using a window of +/- 50Kb around each gene. We used SNP annotations from the 1000 genomes European dataset and gene annotations from the NCBI website build 38."

Line 175-176 (results section):

"We note that the cyan module was not enriched in a MDD (beta=0.04, SE=0.06, P=0.26) or CAD (beta=-0.03, SE=0.06, P=0.68) GWAS."

- Which PGS is being referred to in line 220? Placental? CVD? MDD? This was repeatedly confusing. I think that the authors should consider using the type of PGS as a subscript in the name (PGS_CVD for example).

This is a very good suggestion, in the revised manuscript we explicitly describe each "PGS" upon each use.

- What cytokines were generated (line 250). Detail is very limited here.

This was a misuse of the word "cytokine" on our part and we have revised it to "cord blood molecules" in the revised manuscript. We have also added substantially more detail to the "cord blood analysis" methods section.

Lines 469-481:

"Molecular characterization of cord blood from the GUSTO cohort (sample size of between 194-251 depending on specific molecule analyzed) was conducted in duplicate using commercially available assays. Samples were randomized across plates and internal controls were used to estimate cross-plate variation. Assays with a coefficient of variation exceeding 20% across internal standards were excluded. Molecular profiles were analyzed using 1 of 3 methods: single molecule array (SIMOA; IL6, IL10, TNFa, IFN gamma, IL4), DropArray (MCP1, TSH, insulin, VEGFA, LH, IgE, FSH, glucagon, IP10, Leptin, MIP1a, CRP, C-peptide, IL1RA, IGFBP7, prolactin, MIP1b, growth hormone, IGFBP3, IL12p40, GLP1) and enzyme-linked immunosorbent assay (ELISA; adiponectin, free testosterone, testosterone). **Supplementary Table 9** describes the individual assays. SIMOA measurements were made using the SIMOA HD-1 Analyzer (Quanterix). DropArray measurements were made using the FlexMAP3D bead-based multiplex system (Luminex). Normalization was carried out across plates using a median centring method. Data with readings outside of the assay limits as indicated by the manufacturer were discarded."

- How do you measure homeostatic inflammation (line 258-9)?

Upon reflection on comments from both Reviewers, we realize the term "homeostatic inflammation" was misplaced. As noted in our response to Reviewer 1, we felt compelled to use the term as a reminder to the reader that this was a study of typical development rather than any pathogenic response commonly associated with the use of "inflammation". We now realize that using the term "homeostatic inflammation" is actually unhelpful to the reader. We have reframed the study to focus on "inflammation-related gene expression patterns", "Hofbauer cell function" or simply "the cyan module", as appropriate. We thank the Reviewer for prompting these changes, which we believe have resulted in a far clearer and more concise manuscript.

For example we have changed the title to "Hofbauer cell function in the term placenta is protective against adult cardiovascular and depressive outcomes".

- How did the authors decide on making 3 PGS (lines 296-301)? It seems very limited.

In this section of the manuscript we show that our fetoplacental PGS can specifically predict expression of the cyan module. The three PGS that the Reviewer is referring to, are part of five PGS that we were using as negative controls. We chose the three modules to use as negative controls simply because they were the closest in size to the Cyan module (486 genes): Midnightblue (442 genes), Lightcyan (427 genes) and Salmon (607 genes).

We have now described the rationale for choosing each PGS in the methods section:

Lines 451-457:

"Negative control PGS were also generated to assess the specificity of the fetoplacental PGS to predict cyan module expression. These negative control PGS consisted of the three WGCNA modules closest in size to the cyan module (Midnightblue (442 genes), Lightcyan (427 genes) and Salmon (607 genes)) and two random selections of 486 genes from the sequencing dataset. These negative control PGS were built in an identical fashion to the fetoplacental PGS, using placental eQTLs. As an additional control we used the cyan module to generate another PGS but weighted it based on eQTLs from the fetal cerebral cortex."

- Why is there a range (line 309)? I'm not sure I understand what this means.

We measured 27 different molecules in cord blood using different assays (Supplementary table 9 lists the individual assays used). For each of the 27 molecules we measured the coefficient of variation across technical replicates and excluded those samples that had unacceptable levels of variation. Therefore, even though we start with

the same number of samples different samples are excluded from different assays for technical reasons. We have clarified these points in both the results and methods sections of the revised manuscript:

Line 149-150:

"..cord blood data (n=194-251 depending on molecule measured)"

- How is the association between the fetoplacental PGS and diagnoses different from PheWAS? (line 329)

Our pheWAS was largely absent of any diagnoses and we therefore conducted an analysis of candidate diagnoses based on the pheWAS analysis.

The critical analyses in this section of the manuscript are the pheWAS and the Mendelian randomization results. We now realize the logistic regression results add a layer of confusion for the reader, while being dispensable for both the overall narrative and our conclusions. Therefore, we have removed this analysis of diagnosis from the revised manuscript.

- The risk between inflammation and CVD is confusing. Sometimes, the paper says increased risk and sometimes protective. I think this needs to be clarified throughout.

This is an important point and one we are very grateful to clarify. We have now made the direction of effect explicit throughout the manuscript text and have reformatted the pheWAS plot to reflect the direction of effect (Figure 2A and in Reviewer figure 4). As outlined in our response to the Reviewer's previous comment, we have also removed the diagnoses comparison to create a more streamlined narrative.

Reviewer figure 4 Miami plot of *pheWAS* results for the fetoplacental PGS in the UK Biobank. Each point represents the $-\log P$ -value for the association between the fetoplacental PGS with a particular trait multiplied by the direction of the effect.

- Which effects are sex dependent (line 461)?

We have now clarified this in the revised manuscript:

Line 300:

"We found a female-specific effect of placental eQTLs for cyan module genes on the risk of suicidality."

- What is the ancestry of the summary statistics used for the generation of PGS? Does it match the study samples here?

Yes indeed, the genetic ancestries of the GWAS summary statistics are from individuals of European genetic ancestry and the study samples are also primarily of European genetic ancestry. We also include genetic principal components in all regression analyses to account for population stratification¹². We have now clarified this point in the methods section:

Line 459-460:

"GWAS summary statistics from European ancestry populations by Nikpay *et al* 2015 and Howard *et al*"

Line 526-527:

"In all relevant genetic regression analyses population stratification was accounted for by including genetic principal components as covariates."

Minor

- Methods, line 98, consider adding a reference for UK Biobank

We thank the Reviewer for pointing this out and have added a reference to Bycroft *et al.*

- Why is 23andme excluded? This means that you may need to provide the sumstats files if they are not available. Otherwise, the results are not reproducible.

The Reviewer raises an important point that we are glad to clarify. We used the summary statistics as provided by the psychiatric genomics consortium. The 23 and me sample is excluded by default, as 23 and me require a data access agreement before releasing any summary statistics produced with their data. However, the 23 and me team have been unresponsive to requests for access to data, with data requests from several of our collaborators still unanswered after several years. Therefore, we refer to the summary statistics as "excluding 23 and me" to acknowledge that we have not been given access to the full summary statistics. Even though we phrased it as such for full transparency, we realize it is confusing to the reader and we now refer to the summary statistics as the following:

Line 458-459:

"major depressive disorder (MDD; as provided by the psychiatric genomics consortium)"

- Why is Table 8 the first table mentioned (line 211). Typically, the tables are numbered in the order they appear in the paper.

We have now formatted the manuscript and adjusted all table numbers.

- I believe GSEA is a typo on line 201 where it says GSVA

The method we used (ssGSEA) is confusingly implemented through a package called GSVA. To avoid confusion we refer to the full acronym in the revised manuscript:

Line 463:

"GSVA (Gene Set Variation Analysis)"

- When referring to sex of the samples, I assume these are the offspring (line 261). But it does not say that. It must be since only women give birth, but it is these types of details that I think need to be added.

We now refer to offspring sex and have thoroughly revised the manuscript to aide clarity.

Once more we would like to thank the Reviewer for their time, effort and constructive comments on our manuscript.

REVIEWERS' COMMENTS

Reviewer #1 (Remarks to the Author):

I want to thank the authors for their careful consideration of my comments and willingness to adjust their manuscript. The authors have added increased clarity within their methods and results, and the new presentation of the results is more in alignment with their data. I agree with the authors response to my comment 3 (about using a priori go terms), which is less relevant given their shift in focus of this manuscript. I also wanted to thank the authors for their clarification for the link between the physical and mental health measures and cardiometabolic and mental health disorders. I have a few minor comments to further improve this manuscript.

1. Given the shift in the focus their narrative to a gene signature of Hofbauer cells, I think the way that they are introduced in the introduction should be expanded upon, and they should not be introduced as an example of an immune cell type, but as a more central focus. When the authors describe Hofbauer cells as having "important homeostatic roles", but do not go into detail about these roles. Can the authors provide more detail here with references and expand upon this?

2. When the authors say "Furthermore, loss of immune cells in animal models is sufficient for pregnancy termination."- are they referring to maternal or fetal immune cells? Or both? The authors have added necessary details about inflammation, but from the sentences that have been added it is not entirely clear if they are referring to maternal immune activation or placental/fetal immune activation, so it would improve interpretability to tighten up this language.

3. The presentation of the top 10 enriched GO terms in supplemental table 3 is much more transparent and provides a holistic summary of the data. The newly revised manuscript provides strong evidence in Figure 1B that the cyan module is enriched for Hofbauer cells. It is surprising to me though that none of the modules presented here are associated with trophoblasts (which I would assume would be the most prevalent cell type). Can the authors comment on that?

4. The authors have altered their analysis to using a single cell reference panel of term placental samples, which matches their own timepoint. However, the dataset (Culligan et al, 2023) selected was generated in the context of participants that were actively infected with COVID-19, and the authors of this paper note altered gene expression related to COVID expression. Given the focus on this paper on homeostatic inflammation/immune signaling, I would be cautious about these infected samples as a reference panel. Can the authors justify this, and confirm this with other single-cell RNA sequencing datasets from term placentas, such as recent work published in nature communications (<https://www.nature.com/articles/s42003-023-04623-6>), which studies healthy, term placentas from healthy villous samples and includes other datasets that were also healthy term placental samples for a larger and more comprehensive dataset.

5. Please check the spelling of preeclampsia throughout the manuscript (particularly in Figure 3A)

6. One strength of this study is the diverse population that is often not well represented in omics analyses (Chinese, Malay, Indians). However, these populations are often not well represented in GWAS studies (and in the UK biobank, where 95% of participants are white). How do the authors think this could have influenced their results?

Reviewer #4 (Remarks to the Author):

This is an interesting study where the authors have demonstrated a correlation between Hofbauer cell

gene signature and adult cardiovascular and depressive outcomes . Overall, the manuscript is well written and the data is clearly presented. The findings and methodology used will be of wide interest.

Minor comments:

The title overstates the authors findings. This study is correlative and does not directly show that Hofbauer cell biology is protective against adult cardiovascular and depressive outcomes.

Line 40-42: I think the wording used here is not clear and the sentence should be rephrased. Hofbauer cells do not have a function after birth. Given that, this sentence does not make sense. I suggest switching the 'following' with 'due to'.

Line 62 and 63: A reference is not provided for the origin of microglia and Hofbauer cells. Study demonstrating microglia origin: DOI: [10.1016/j.immuni.2015.03.011](https://doi.org/10.1016/j.immuni.2015.03.011)
More recent studies in mice (DOI: [10.1016/j.devcel.2021.06.005](https://doi.org/10.1016/j.devcel.2021.06.005)) and humans (DOI: [10.1038/s41467-023-37383-2](https://doi.org/10.1038/s41467-023-37383-2)) suggest Hofbauer cells are generated de novo within the placenta and are not of yolk sac origin.

Reviewer #1:

I want to thank the authors for their careful consideration of my comments and willingness to adjust their manuscript. The authors have added increased clarity within their methods and results, and the new presentation of the results is more in alignment with their data. I agree with the authors response to my comment 3 (about using a priori go terms), which is less relevant given their shift in focus of this manuscript. I also wanted to thank the authors for their clarification for the link between the physical and mental health measures and cardiometabolic and mental health disorders. I have a few minor comments to further improve this manuscript.

We thank the Reviewer again for their time and constructive comments on our manuscript. We have considered each of their points in detail below.

1. Given the shift in the focus their narrative to a gene signature of hofbauer cells, I think the way that they are introduced in the introduction should be expanded upon, and they should not be introduced as an example of an immune cell type, but as a more central focus . When the authors describe Hoffbauer cells as having “important homeostatic roles”, but do not go into detail about these roles. Can the authors provide more detail here with references and expand upon this?

We appreciate the opportunity to refine our narrative and, as suggested by the Reviewer, we have now elaborated on Hofbauer cell function in the introduction.

Lines 62-67

“For instance, microglia are a tissue-specific macrophage and the placenta is home to its own tissue-specific macrophage, Hofbauer cells, which have important homeostatic roles in angiogenesis, placental remodelling and trophoblast maturation ¹⁻³, while also displaying a robust response to infection ¹. However, little work has been done to identify the contribution of Hofbauer cells to adult health outcomes under non-pathogenic conditions.”

2. When the authors say “Furthermore, loss of immune cells in animal models is sufficient for pregnancy termination.”- are they referring to maternal or fetal immune cells? Or both? The authors have added necessary details about inflammation, but from the sentences that have been added it is not entirely clear if they are referring to maternal immune activation or placental/fetal immune activation, so it would improve interpretability to tighten up this language.

We thank the Reviewer for noting this point. We have now made this sentence more specific.

Lines 58-60

“Furthermore, the loss of maternal decidual natural killer cells ⁴ or uterine dendritic cells ⁵ is sufficient for pregnancy termination in animal models.”

3. The presentation of the top 10 enriched GO terms in supplemental table 3 is much more transparent and provides a holistic summary of the data. The newly revised manuscript provides strong evidence in Figure 1B that the cyan module is enriched for Hoffbauer cells. It is surprising to me though that none of the modules presented here are associated with trophoblasts (which I would assume would be the most prevalent cell type). Can the authors comment on that?

We agree with the Reviewer the lack of expression in trophoblasts is somewhat surprising. We expect this is because we only characterized modules prominently associated with inflammation. We explicitly tested this by creating scores for each of the WGCNA modules, once more using the control, term samples only from Lu Culligan *et al.* We present this data below in Reviewer Figure 1. We observe several modules that are highly expressed in trophoblasts, which, as noted by the Reviewer, is expected. We also note that we provide a list of all of our WGCNA modules in the Supplementary Data 2 for readers to utilize in their own analyses.

Reviewer Figure 1 Module scores in healthy term placental villus samples (Lu Culligan *et al.*) for all 28 WGCNA modules we identified in our bulk RNA-seq dataset

4. The authors have altered their analysis to using a single cell reference panel of term placental samples, which matches their own timepoint. However, the dataset (Culligan *et al.*, 2023) selected was generated in the context of participants that were actively infected with COVID-19, and the authors of this paper note altered gene expression related to COVID expression. Given the focus on this paper on homeostatic inflammation/immune signaling, I would be cautious about these infected samples as a reference panel. Can the authors justify this, and/confirm this with other single-cell RNA sequencing datasets from term placentas, such as recent work published in nature communications (<https://www.nature.com/articles/s42003-023-04623-6>), which studies healthy, term placentas from healthy villous samples and includes other datasets that were also healthy term placental samples for a larger and more comprehensive dataset.

The Reviewer raises an important point, which we are very happy to clarify. With respect to the Lu Culligan *et al.* dataset, we used only the control samples to test cell type expression. We did this to avoid COVID-related expression artefacts, as also noted by the Reviewer. We have now made this point explicitly clear in the text of the revised manuscript.

However, we thank the Reviewer for pointing us to the Campbell *et al.* dataset and appreciate the opportunity to further validate our findings. We used their data and cell type annotations to once more test the specificity of the cyan module for Hofbauer cells. We present these results below in

Reviewer Figure 2 and in Supplementary Fig 2a of the revised manuscript, where we again see notable specificity of the cyan module for Hofbauer cells.

Line 107-109

“ We then built expression scores for each of the inflammation-related modules in a scRNA-seq dataset of term placental villous tissue (using only healthy control samples) ^{6,7} (Figure 1b and Supplementary Fig 2a).”

Reviewer Figure 2 Module scores for the five inflammation-related modules in the Campbell et al scRNA-seq dataset

5. Please check the spelling of preeclampsia throughout the manuscript (particularly in Figure 3A)

We thank the Reviewer for noticing this error and have now corrected the spelling in Figure 3a and have further proofread the manuscript.

6. One strength of this study is the diverse population that is often not well represented in omics analyses (Chinese, Malay, Indians). However, these populations are often not well represented in GWAS studies (and in the UK biobank, where 95% of participants are white). How do the authors think this could have influenced their results?

As the Reviewer notes there are several necessary considerations when using diverse genetic ancestries in genomic studies. We tried to minimize these concerns by using eQTLs identified in the Rhode Island Child Health Study, which has a similar distribution of genetic ancestries to the UK Biobank (89.3% white European genetically inferred ancestry⁸). Therefore, while our RNA-seq data was indeed derived from non-European samples, the genetic scores (which as the Reviewer notes are sensitive to genetic ancestry) were generated using data from genetic ancestries comparable to the UK Biobank. Furthermore, we also used genetic principal components in our analysis to account for population stratification. While in our Mendelian randomization analyses, we used GWAS conducted in populations of the same genetic ancestries.

However, we share the Reviewer's enthusiasm for further genomic studies in diverse populations and believe these studies will play an important role in advancing science. We have now noted this sentiment in the discussion section of our revised manuscript.

Lines 317-319:

"Future work armed with larger and more genetically diverse eQTL databases that examine an expanded catalogue of diagnoses will be well placed to expand upon our results."

Reviewer #4 (Remarks to the Author):

This is an interesting study where the authors have demonstrated a correlation between Hofbauer cell gene signature and adult cardiovascular and depressive outcomes . Overall, the manuscript is well written and the data is clearly presented. The findings and methodology used will be of wide interest.

We thank the Reviewer for their time and constructive feedback on our manuscript. We address each of their comments below.

Minor comments:

The title overstates the authors findings. This study is correlative and does not directly show that Hofbauer cell biology is protective against adult cardiovascular and depressive outcomes.

We have changed the title of the revised manuscript in line with the Reviewer's comment to:

"Hofbauer cell function in the term placenta associates with adult cardiovascular and depressive outcomes"

Line 40-42: I think the wording used here is not clear and the sentence should be rephrased. Hofbauer cells do not have a function after birth. Given that, this sentence does not make sense. I suggest switching the 'following' with 'due to'.

We thank the Reviewer for noting this error and have changed the referenced wording.

Line 40-42

"Our data support a model where disruption of placental Hofbauer function, due to preterm birth or prenatal infection, contributes to the increased risk of depression and cardiovascular disease observed in these individuals."

Line 62 and 63: A reference is not provided for the origin of microglia and Hofbauer cells.

Study demonstrating microglia origin: DOI: 10.1016/j.immuni.2015.03.011

More recent studies in mice (DOI: 10.1016/j.devcel.2021.06.005) and humans (DOI:

10.1038/s41467-023-37383-2) suggest Hofbauer cells are generated de novo within the placenta and are not of yolk sac origin.

We agree with the Reviewer, the definitive origin of Hofbauer cells is contested in the field and a reference to their origin adds little to the narrative of our manuscript. Therefore, we have removed this comparison and instead now refer to both Hofbauer cells and microglia as tissue-specific macrophages.

Lines 62-63 of the introduction:

"For instance, microglia are tissue-specific macrophages and the placenta is home to its own tissue-specific macrophage, Hofbauer cells"

References

1. Thomas, J. R. *et al.* Phenotypic and functional characterization of first-trimester human placental macrophages, Hofbauer cells. *J. Exp. Med.* **218**, (2020).
2. Reyes, L. & Golos, T. G. Hofbauer cells: Their role in healthy and complicated pregnancy. *Front. Immunol.* **9**, 2628 (2018).
3. Khan, S., Katabuchi, H., Araki, M., Nishimura, R. & Okamura, H. Human Villous Macrophage-Conditioned Media Enhance Human Trophoblast Growth and Differentiation In Vitro. *Biol. Reprod.* **62**, 1075–1083 (2000).
4. Hanna, J. *et al.* Decidual NK cells regulate key developmental processes at the human fetal-maternal interface. *Nat. Med.* **2006 129 12**, 1065–1074 (2006).
5. Plaks, V. *et al.* Uterine DCs are crucial for decidua formation during embryo implantation in mice. *J. Clin. Invest.* **118**, 3954–3965 (2008).
6. Lu-Culligan, A. *et al.* Maternal respiratory SARS-CoV-2 infection in pregnancy is associated with a robust inflammatory response at the maternal-fetal interface. *Med* **2**, 591-610.e10 (2021).
7. Campbell, K. A. *et al.* Placental cell type deconvolution reveals that cell proportions drive preeclampsia gene expression differences. *Commun. Biol.* **2023 61 6**, 1–15 (2023).
8. Peng, S. *et al.* Expression quantitative trait loci (eQTLs) in human placentas suggest developmental origins of complex diseases. *Hum. Mol. Genet.* **26**, 3432 (2017).